# Individual tree point clouds and tree measurements from multi-platform laser scanning in German forests

Hannah Weiser[1], Jannika Schäfer[2], Lukas Winiwarter[1], Nina Krašovec[1,3], Fabian E. Fassnacht[2,4], and Bernhard Höfle[1,5]

[1]3DGeo Research Group, Institute of Geography, Heidelberg University, Germany
[2]Institute of Geography and Geoecology, Karlsruhe Institute of Technology, Karlsruhe, Germany
[3]Department of Psychiatry and Psychotherapy, Central Institute of Mental Health, Medical Faculty Mannheim, Heidelberg University, Mannheim, Germany
[4]Remote Sensing and Geoinformatics, Freie Universität Berlin, Berlin, Germany
[5]Interdisciplinary Center for Scientific Computing (IWR), Heidelberg University, Germany

**Correspondence:** Hannah Weiser (h.weiser@uni-heidelberg.de)

**Abstract.** Laser scanning from different acquisition platforms enables collecting 3D point clouds from different perspectives and with varying resolutions. These point clouds allow us to retrieve detailed information on the individual tree and forest structure. We conducted airborne laser scanning (ALS), uncrewed aerial vehicle (UAV)-borne laser scanning (ULS) and terrestrial laser scanning (TLS) in two German mixed forests with species typical for Central Europe. We provide the spatially overlapping, georeferenced point clouds for 12 forest plots. As a result of individual tree extraction, we furthermore present a comprehensive database of tree point clouds and corresponding tree metrics. Tree metrics were derived from the point clouds and, for half of the plots, also measured in the field. Our dataset may be used for the creation of 3D tree models for radiative transfer modeling, LiDAR simulation studies or to fit allometric equations between point cloud metrics and forest inventory variables. It can further serve as a benchmark dataset for different algorithms and machine learning tasks, in particular automated individual tree segmentation, tree species classification or forest inventory metric prediction. The dataset and supplementary metadata are available for download, hosted by the PANGAEA data publisher at https://doi.org/10.1594/PANGAEA.942856 (Weiser et al., 2022b).

## 1 Introduction

Airborne and terrestrial laser scanning enables the collection of detailed 3D information on forest structure. This information can be used to estimate important tree and forest stand attributes including tree and stand height (Pearse et al., 2019; Wang et al., 2019), wood volume (Dassot et al., 2012; Disney et al., 2018), biomass (Disney et al., 2018; Næsset et al., 2011), canopy cover (Arumäe and Lang, 2018; Smith et al., 2009), tree density (Pearse et al., 2019), diameter at breast height (DBH) (Bruggisser et al., 2020; Liu et al., 2018) and basal area (Pearse et al., 2019). Laser scanning data from airplanes and helicopters have been widely applied in the forest sector and are operationally used for forest inventories in several countries (Montaghi et al., 2013; Scion, 2021; White et al., 2017) using the area-based approach (Bouvier et al., 2015; Holopainen et al., 2014; Næsset, 2002). In the area-based approach, wall-to-wall laser scanning data is combined with information collected in field plots to estimate

forest attributes across the complete area covered by the laser scanning survey. Besides the area-based approach, individual tree-based approaches in which first single trees are detected from the point clouds and then attributes are estimated for the individual trees have also received attention (Latifi et al., 2015; Maltamo et al., 2004). Terrestrial laser scanning data with stationary or mobile sensors can be used to obtain detailed information on individual trees which typically cannot be directly derived from airborne surveys. Corresponding attributes include DBH and tree stem taper information but also information on small-scale understorey variation which may be important for characterizing local fuel conditions or habitat properties. Finally, laser scanning with uncrewed aerial vehicles (UAVs) represents a compromise between airborne and terrestrial laser scanning. UAV-borne laser scanning (ULS) allows for data collection over larger areas compared to TLS but cannot reach the fine spatial resolution of TLS data. Due to the lower flying height, ULS achieves higher point densities and a better penetration of the canopy than ALS, but covers smaller areas due to constraints in flying time and speed.

Over the last years, notable progress has been made in both sensor systems and workflows to derive forest attributes from laser scanning data collected by various platforms (Calders et al., 2020; Maltamo et al., 2014; Morsdorf et al., 2017; Roussel et al., 2020; Wallace et al., 2014). At the same time, there is still space for improvements for example when it comes to (1) understanding and comparing the potential of tree delineation algorithms or (2) deriving certain attributes such as tree species from laser scanning point clouds. Furthermore, (3) there are still open questions related to the influence of acquisition settings on the model performances in the area-based approach. For issues (1) and (2), obtaining reliable reference data in the field is often extremely challenging and time-intensive and preparing openly-available high-quality benchmark datasets may contribute to make further advancement in the research field. For issue (3), cost is often a limiting factor since in many cases it is impossible to acquire numerous ALS flights in a research project to examine the influence of acquisition parameters due to the high costs of the flights. In this case, a combination of synthetic forest stands and laser scanning simulators may be one way forward by enabling extensive sensitivity analyses (Disney et al., 2010; Roberts et al., 2020).

One key-element for creating synthetic forest stands, enabling laser scanning simulations, are detailed and realistic 3D tree models, which can be reconstructed from 3D point clouds obtained with terrestrial, UAV-borne or airborne laser scanning. Virtual forest scenes can be built from such 3D tree models and used as input to complex radiative transfer models (e.g., DART, Gastellu-Etchegorry et al. 2015) or Light Detection and Ranging (LiDAR) simulators (e.g., HELIOS++, Winiwarter et al. 2022). Such simulations enable an improved understanding of the relation between forests' structural elements and electromagnetic radiation (Disney et al., 2010; Weiser et al., 2021; Widlowski et al., 2015). This is particularly relevant for improving remote sensing-assisted forest inventory approaches. 3D tree models derived from laser scans of real trees have also been used to develop structural tree allometries to, e.g., non-destructively estimate the above-ground biomass of tree individuals (Calders et al., 2015). Further, 3D tree models can be used to parameterize eco-physiological models (Sinoquet et al., 2001) to quantify shading effects and evapotranspiration properties of urban trees. Other applications of detailed tree models include their integration in 3D visualizations as, e.g., used in urban landscape planning projects or in computer games (Bournez et al., 2017). (Semi-)automated methods to extract tree models from TLS point clouds are readily available (e.g., SimpleTree/SimpleForest, Hackenberg et al. 2015; TreeQSM, Raumonen et al. 2013) but data collection and extraction of high-quality tree point clouds is still very time-consuming because it requires manual interaction and control. This generally limits the acquisition and pro-

cessing of very large numbers of trees in single projects and thus underlines the importance of publishing such datasets as open data for shared usage.

The dataset presented here is one of the biggest open-access collections of multi-platform tree point clouds with matching inventory data of common tree species of Central European forests. We present 1491 trees which were identified in point clouds acquired in 12 forest plots in southwest Germany. 249 trees were extracted from all three types of point clouds, ALS, ULS and TLS. Another 1031 trees were extracted from both ALS and ULS point clouds. 1168 trees were extracted from two different ULS datasets, acquired under leaf-on conditions and leaf-off conditions (Figure 8). The maximum time period between laser scanning acquisitions under leaf-on conditions is 70 days. Single tree metrics, including species, height, DBH and crown metrics, were derived from the point clouds and, for half of the plots (1060 trees), also measured in the field. Thus, one or more point clouds and corresponding metrics are provided for each tree. In addition to the 3D coordinates, each point of the point cloud has attributes including reflectance, echo number, waveform deviation and time of recording. The dataset further exists of the larger area forest point clouds of the acquisitions and the flight trajectory or scan position information.

Figure 1 displays the workflow of the data acquisition and processing steps.

## 2 Methods

In this section, we introduce the study site, give an overview of the measurements taken in the field and document the LiDAR acquisitions from the three different platforms (airborne, UAV-borne, terrestrial) which were conducted in 2019 and 2020 under leaf-on and leaf-off canopy conditions. We describe how we extracted single tree point clouds from the LiDAR datasets to a) match them to the field measurements, and b) derive individual tree metrics directly from the point clouds.

## 2.1 Study site

The two study study sites, the Bretten municipal forest (49°00'36"N 8°41'35"E) and the Hardtwald forest in Karlsruhe-Waldstadt (49°02'04"N 8°25'40"E), are located in a region of temperate forests in the federal state of Baden-Württemberg, Germany (Figure 2). Data acquisitions cover three forest plots in the Karlsruhe forest and nine in the Bretten municipal forest, totaling twelve forest plots (Figure 2). The Bretten site is characterized by the hilly landscape of the Kraichgau, and the Karlsruhe site is on flat terrain of the Upper Rhine Plain. The main tree species in the managed forest stands are Scots pine (*Pinus sylvestris* L.), European beech (*Fagus sylvatica* L.), common oak (*Quercus robur* L.), sessile oak (*Quercus petraea* (Matt.) Liebl.), red oak (*Quercus rubra* L.), European hornbeam (*Carpinus betulus* L.), Douglas fir (*Pseudotsuga menziesii* (Mirb.) Franco), and Norway spruce (*Picea abies* (L.) H. Karst). Most stands have multiple layers, a mixed species composition, and a dense forest cover. Karlsruhe plots are dominated by Scots pine, red oak and beech. Bretten plots are more diverse and characterized by beech, spruce, Douglas fir, oaks (mostly sessile oak) and European hornbeam. We provide a summary of forest plot characteristics in Table A1, including the number of trees, basal area $[\mathrm{m^2\,ha^{-1}}]$, the percentage of dead trees and the median tree height $[\mathrm{m}]$, also separated by tree species.

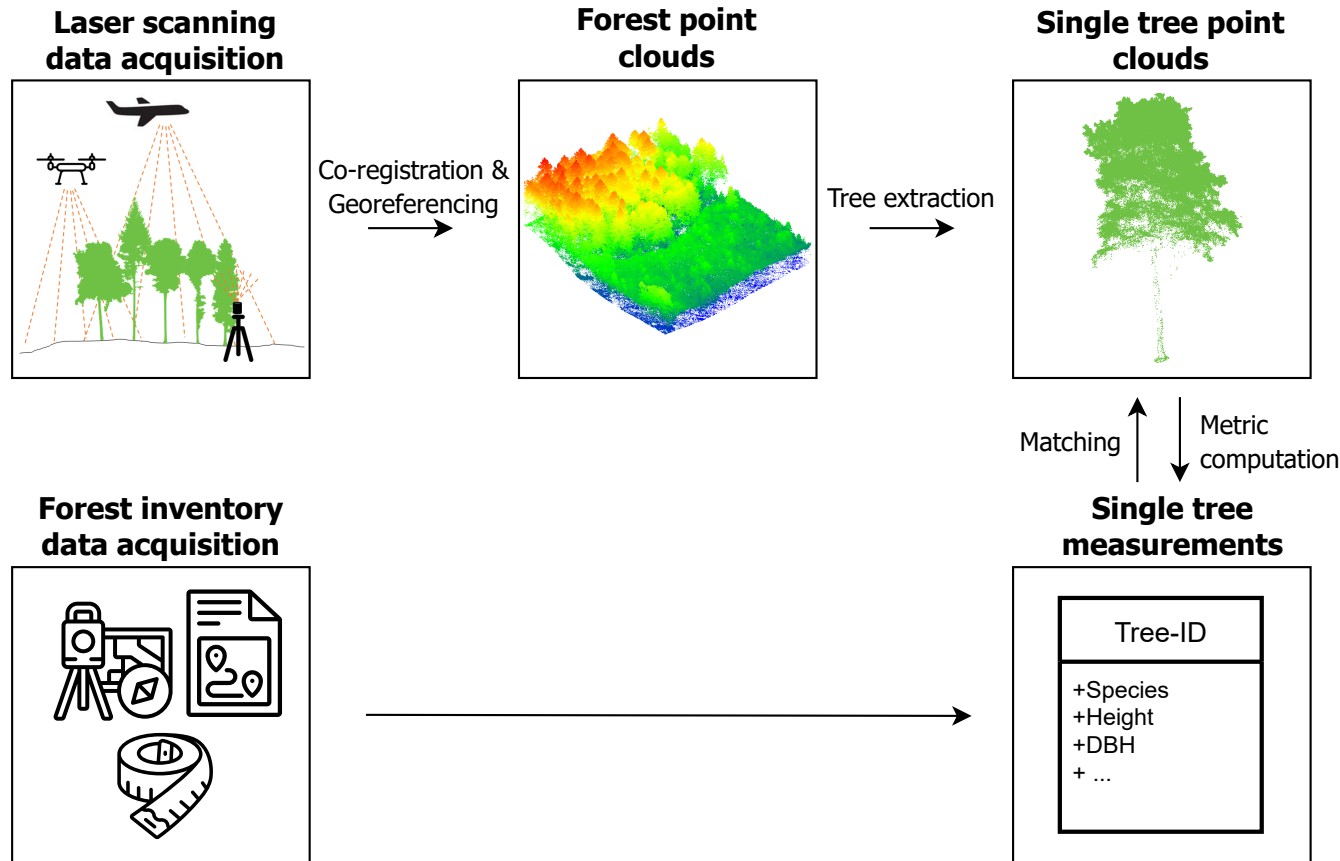

**Figure 1.** Workflow for the generation of the dataset, starting with the laser scanning data acquisition and forest inventory measurements and resulting in single tree point clouds and corresponding point cloud-derived and field-measured metrics. This figure has been designed using resources from http://www.flaticon.com/ (Smashicons, Freepik).

## 2.2 Laser scanning data acquisition and co-registration of the multi-modal datasets

### 2.2.1 Airborne laser scanning (ALS)

MILAN Geoservice GmbH was commissioned by the Institute of Geography and Geoecology (IFGG) of Karlsruhe Institute of Technology (KIT) to carry out the ALS acquisitions. The flights were conducted on July 5, 2019 under leaf-on conditions with a *RIEGL* VQ780i (RIEGL Laser Measurement Systems, 2019) sensor mounted on an aircraft of type Cessna C207. Data were acquired in 38 parallel flight strips, plus two orthogonal strips, in Karlsruhe, covering an area of around $65\,\mathrm{km}^2$. In Bretten, data were acquired in 25 parallel flight strips, plus two orthogonal strips, covering an area of around $32\,\mathrm{km}^2$. Detailed sensor

and acquisition parameters are listed in Table 1. The data were georeferenced and transformed into the coordinate reference

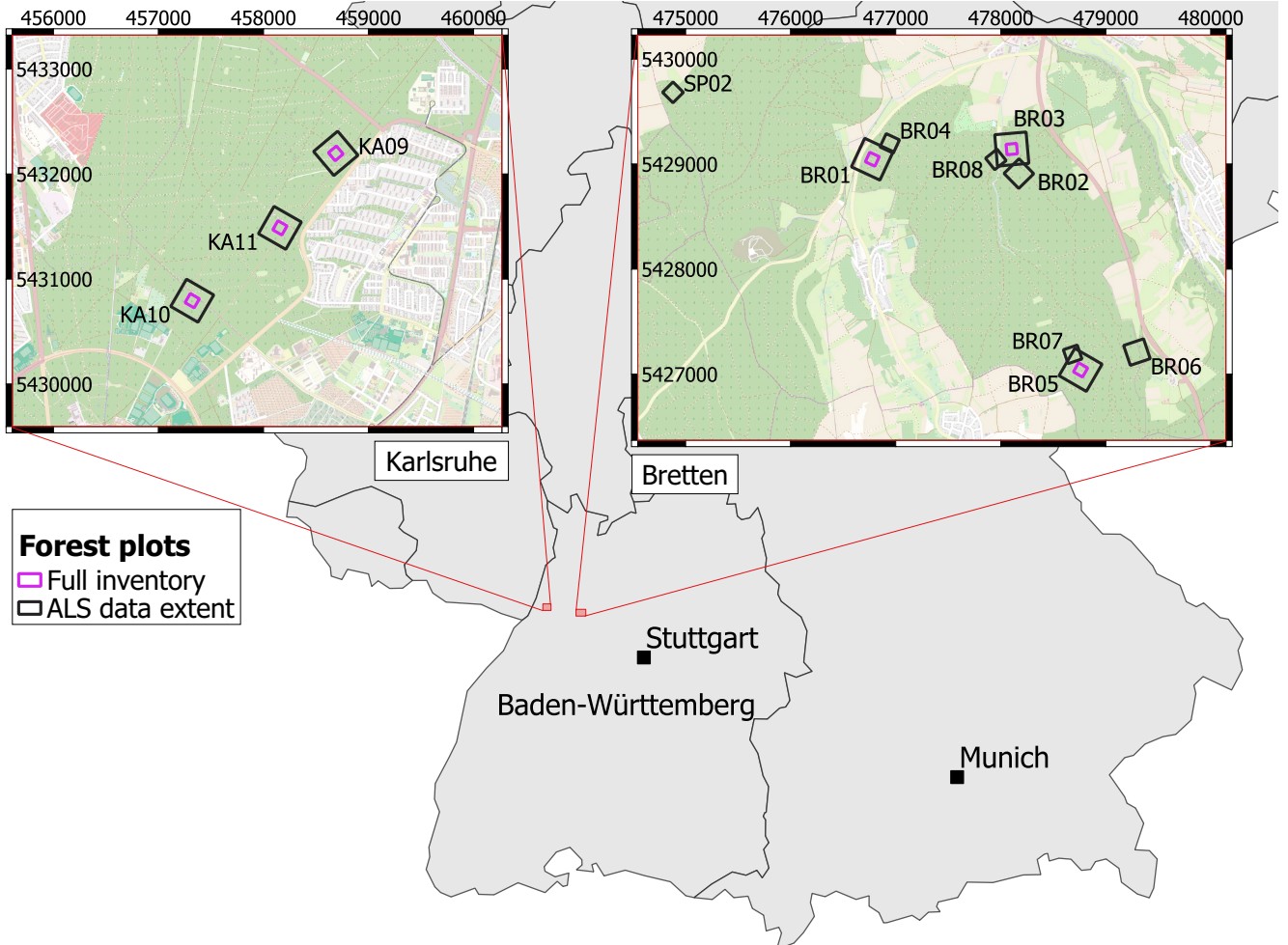

**Figure 2.** Map of the forest plots in Karlsruhe (left) and Bretten (right) in the federal state of Baden-Württemberg, Germany. The extent of the extracted ALS data is outlined in black, areas for which field measurements were conducted are outlined in purple ("Full inventory"). All plots except BR04 are covered by the ULS acquisitions. TLS was performed in selected locations within the plots. Coordinates are given in the projected coordinate system ETRS89 / UTM zone 32N (EPSG:25832). Map data: © OpenStreetMap contributors 2021. Distributed under the Open Data Commons Open Database License (ODbL) v1.0. Administrative boundaries: © EuroGeographics (2021).

system ETRS89 / UTM zone 32 (EPSG:25832) with ellipsoidal height (GRS80). Full-waveform data are available from the authors on reasonable request.

**Table 1.** Airborne laser scanning (ALS), UAV-borne laser scanning (ULS) and terrestrial laser scanning (TLS) sensor and acquisition parameters. Sensor specifications are according to the datasheets (RIEGL Laser Measurement Systems, 2017, 2019, 2020b)

| Mode | ALS |
|---|---|
| Aircraft | Cessna C207 |
| Sensor | *RIEGL* VQ-780i |
| Laser beam divergence | 0.25 mrad (measured at the $1/e^2$ points) |
| Pulse repetition frequency | 1000 kHz |
| Scan frequency | 225 lines per second |
| Altitude above ground | 650 m |
| Off-nadir scan angle | $\pm30°$ |
| Flight line distance | 175 m |
| Flight line overlap | 76 % |
| Speed | 100 kn ($\approx 51.4 \, \mathrm{m\,s^{-1}}$) |
| Date | 2019-07-05 |
| **Mode** | **ULS** |
| UAV | DJI Matrice 600 Pro |
| Sensor | *RIEGL* miniVUX-1UAV |
| Laser beam divergence | $\approx 2.72$ mrad $\times$ 0.85 mrad (measured at the $1/e^2$ points; 1.6 mrad $\times$ 0.5 mrad measured at 50 % peak intensity) |
| Pulse repetition frequency | 100 kHz |
| Off-nadir scan angle | $\pm90°$ |
| **Mode** | **TLS** |
| Sensor | *RIEGL* VZ-400 |
| Laser beam divergence | 0.3 mrad (measured at the $1/e^2$ points) |
| Pulse repetition frequency | 300 kHz |
| Scan angle range | 100° (+60°/−40° from the horizontal plane) |

For quality control, seven small built-up areas surrounding the area of interest were recorded by the 3DGeo Research Group of Heidelberg University using ULS, for which georeferencing was carried out as described in the next section.

Flight strips were merged and the point clouds were clipped to the 12 forest plots, using a generously sized buffer to cover the other datasets (ULS, TLS, field measurements). This resulted in extracted ALS point clouds of 300 m×300 m, 200 m×200 m, or 120 m×120 m extent (Figure 2), depending on the spatial coverage of the TLS and ULS datasets. Pulse density grids with

a cell size of $1\,\mathrm{m}$ were computed for each point cloud. Pulse density, averaged over all 12 plot point clouds, was $72.5\,\mathrm{pts/m^2}$ (standard deviation: $31.0\,\mathrm{pts/m^2}$).

### 2.2.2 UAV-borne laser scanning (ULS)

UAV-borne laser scanning point clouds were acquired with a *RIEGL* miniVUX-1UAV (RIEGL Laser Measurement Systems, 2020b) sensor mounted on a DJI Matrice 600 Pro Hexacopter (DJI, 2018). Flights were conducted in August and September 2019, in December 2019, and in March and April 2020 to obtain data of trees under both leaf-on and leaf-off conditions. Table 1 shows the general settings used in all 21 flights. For plots KA10 and KA11 in Karlsruhe, only leaf-on data were acquired in 110 summer. Two overlapping double grid patterns, offset by $45°$ in orientation were employed for all flights (Figure 3). In the 2019 flights, adaptive banked turns were flown, while in the 2020 flights, the UAV was operated in stop and turn-mode, i.e., stopped in each corner to turn without inclining about the roll axis. Flying altitude, speed and flight line distance varied across the different flight campaigns and are listed in Table 2. Note that for the 2019 flights of plots BR03 and BR08, the two parts of the double cross were flown at different speeds.

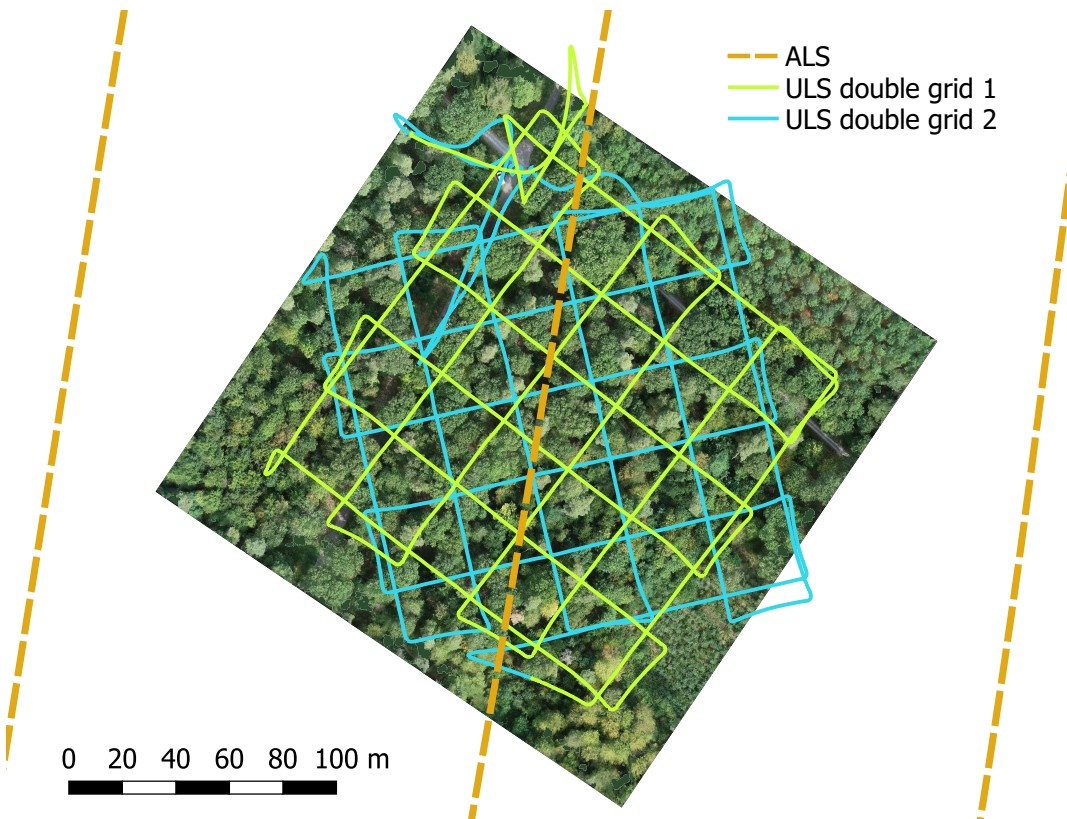

**Figure 3.** UAV-borne laser scanning (ULS) and airborne laser scanning (ALS) flight trajectories of plot KA10. Background: Orthophoto of the forest plot, acquired on 2019-09-06.

**Table 2.** UAV-borne laser scanning acquisition parameters per flight and resulting approximate pulse densities, quantified on extracted point clouds of 1 ha, thereby excluding points on the flight line edges.

| Plot | Date | Canopy condition | Scan frequency [lines/s] | Altitude above ground [m] | Speed [m s$^{-1}$] | Flight line distance [m] | Mean pulse density [pts m$^{-2}$] |
|---|---|---|---|---|---|---|---|
| BR01 | 2019-09-12 | leaf-on | 33.4 | 68.3, $\pm$ 2.4 | 5.0 | 30 | 936 |
| | 2019-12-04 | leaf-off | 33.0 | 62.3, $\pm$ 1.7 | 5.0 | 30 | 859 |
| | 2020-03-27 | leaf-off | 33.4 | 72.4, $\pm$ 1.9 | 5.0 | 27 | 824 |
| BR02 | 2019-08-24 | leaf-on | 100 | 72.1, $\pm$ 3.4 | 5.0 | 26 | 1091 |
| | 2020-04-01 | leaf-off | 33.4 | 62.4, $\pm$ 3.9 | 5.0 | 26 | 1026 |
| BR03 | 2019-08-24 | leaf-on | 100 | 62.5, $\pm$ 1.3 | 5.0 / 4.0 | 26 | 1262 |
| | 2020-03-31 | leaf-off | 33.4 | 67.7, $\pm$ 2.9 | 5.0 | 27 | 1114 |
| BR05 | 2019-09-12 | leaf-on | 33.4 | 70.8, $\pm$ 4.7 | 5.0 | 30 | 909 |
| | 2020-03-26 | leaf-off | 33.4 | 74.5, $\pm$ 5.0 | 5.0 | 27 | 988 |
| BR06 | 2019-09-12 | leaf-on | 33.4 | 61.2, $\pm$ 2.8 | 5.0 | 30 | 953 |
| | 2020-03-27 | leaf-off | 33.4 | 61.2, $\pm$ 3.3 | 5.0 | 26 | 1023 |
| BR07 | 2019-08-24 | leaf-on | 100 | 70.1, $\pm$ 5.7 | 4.4 | 26 | 1115 |
| | 2020-03-26 | leaf-off | 33.4 | 71.4, $\pm$ 5.4 | 5.0 | 27 | 1050 |
| BR08 | 2019-08-24 | leaf-on | 100 | 63.9, $\pm$ 2.2 | 5.0 / 4.0 | 26 | 1205 |
| | 2020-03-31 | leaf-off | 33.4 | 69.3, $\pm$ 2.3 | 5.0 | 27 | 1024 |
| SP02 | 2019-09-03 | leaf-on | 100 | 64.0, $\pm$ 2.5 | 4.0 | 25 | 797 |
| | 2020-04-01 | leaf-off | 33.4 | 66.2, $\pm$ 3.1 | 5.0 | 26 | 919 |
| KA09 | 2019-09-13 | leaf-on | 33.4 | 65.6, $\pm$ 0.8 | 5.0 | 30 | 941 |
| | 2019-12-10 | leaf-off | 33.4 | 60.9, $\pm$ 0.8 | 5.0 | 30 | 987 |
| KA10 | 2019-09-13 | leaf-on | 33.4 | 64.9, $\pm$ 0.8 | 4.8 | 30 | 1036 |
| KA11 | 2019-09-06 | leaf-on | 33.4 | 64.4, $\pm$ 1.0 | 3.0 | 25 | 1554 |

ULS data was processed using the software RiProcess (RIEGL Laser Measurement Systems 2020a, version 1.4.2). GNSS reference measurements were retrieved as a virtual base station from SAPOS Baden-Württemberg (https://www.sapos-bw.de), and were used to post-process the trajectory using Applanix PosPac MMS 8.3 (Applanix Corporation, 2018). The trajectory and the point cloud data from the scanner were imported to RiProcess. At import, the point cloud data were filtered for reflectance between $-22$ dB and $10$ dB, scan angle $\leq \pm 90°$ off-nadir, and ranges between $5$ m and $150$ m to filter outliers from the

data. The point cloud data were converted from the scanner (SDCImport) and then transformed from local polar to world coordinates (ETRS89/UTM Zone 32N) using the flight trajectory (RiWorld). For the subsequent strip adjustment, only points with a scan angle $\leq \pm 45°$ off-nadir were considered, because larger off-nadir angles constitute almost only of within-canopy returns which we presumed to be less accurate and/or less suitable for strip adjustment. Finally, each flight strip (with the full scan angle range of $\pm 90°$) was exported in LAZ format including intensity information (signal amplitude and reflectance) and

waveform information (pulse shape deviation). Using LAStools (Rapidlasso GmbH 2020, version 200509), the single flight strips were merged and the information on the flight lines was encoded in the LAS field "Point Source ID".

Pulse density grids were computed for extracted 1 ha point clouds of the 12 plots (i.e., excluding the edges of the flight lines) with a cell size of $1\,\text{m}$. The mean pulse densities derived for these grids are shown in Table 2.

### 2.2.3   Terrestrial laser scanning (TLS)

TLS acquisitions were performed at selected spots within the 12 forest plots. Our focus was on the most common species in our study site. In total, we conducted 31 TLS surveys on 20 days between June and September 2019 under leaf-on conditions. In each survey, a group of selected trees was scanned from five to eight scan positions (SPs), regularly distributed around the target trees. Depending on the size of the scan area, the terrain, and the forest density, the number and the exact positions of the SPs were adjusted. The scans were acquired with a *RIEGL* VZ-400 terrestrial laser scanner, which has a wavelength of $1550\,\text{nm}$ and

a beam divergence of $0.3\,\text{mrad}$ (full angle, measured at the $1/\text{e}^2$ points; RIEGL Laser Measurement Systems 2017, Table 1). Scans were performed with a pulse repetition frequency (PRF) of $300\,\text{kHz}$ and vertical and horizontal angle increments of $0.017°$ ($0.3\,\text{mrad}$). This results in $3\,\text{mm}$ point spacing at $10\,\text{m}$ range for a single scan and ensures non-overlapping footprints. Only in three acquisitions on June 3, 2019 and June 4, 2019, the vertical resolution was erroneously set to $0.099°$ ($1.73\,\text{mrad}$). Each scan had a vertical field of view of $100°$ ($+60°/-40°$ from the horizontal plane) and a horizontal field of view of $60°$ to

$100°$ facing the target trees. If the distance between scanner and target trees was low in relation to the tree height, we performed an additional tilted scan at respective SPs to increase the vertical extent of the scan. Five cylindrical reflective targets mounted on tripods were positioned in the area to be scanned. They serve as tiepoints in the tiepoint-based co-registration of multiple scans. Further circular reflectors were pinned onto trees of interest to later facilitate their identification in the point cloud.

For each campaign, RTK GNSS measurements from one scan position, referred to as the *main stable position*, and one

tiepoint were used for initial georeferencing of the respective scan. Remaining scans were registered to the main stable position in the software RiSCAN Pro (versions 2.8.0, 2.8.2, 2.9.0) using the cylindrical and circular reflective targets in the scene. After the tiepoint-based registration, a fine alignment was performed using the Multi Station Adjustment (MSA) function of RiSCAN Pro, which uses a variant of the iterative closest point (ICP) algorithm based on planar areas in the point cloud. Because of the weak GNSS signal below the leaf canopy, the positional measurements taken in the forest only allowed coarse georeferencing

of the TLS point clouds. Therefore, TLS point clouds were registered to the ULS point clouds by tree matching and subsequent ICP-alignment. Because stems are sampled better in the leaf-off point clouds due to the higher penetrability of the defoliated crowns, we used the ULS leaf-off point clouds for stem detection where available. TLS point clouds were downsampled for the stem detection using the RiSCAN octree filter with a voxel size of $0.05\,\text{m}$. We calculated normal vectors, linearity and 3D point

densities in corresponding TLS and ULS point clouds using OPALS (Pfeifer et al. 2014, version 2.3.1). By filtering for these three attributes, stem points were extracted and then segmented. To derive the final stem positions, each segment was projected to the horizontal plane and the center of gravity was computed. For each pair of overlapping point clouds, TLS-derived tree positions were manually matched to ULS-derived tree positions and the resulting rigid transformation was applied to the TLS point cloud. The height offset was estimated by selecting corresponding points in a vertical profile in both point clouds. After correcting the height offset, a fine alignment between TLS and ULS point clouds was performed with the ICP implementation of OPALS. A visual assessment of the alignment between ALS, ULS and TLS is presented in Figure 4.

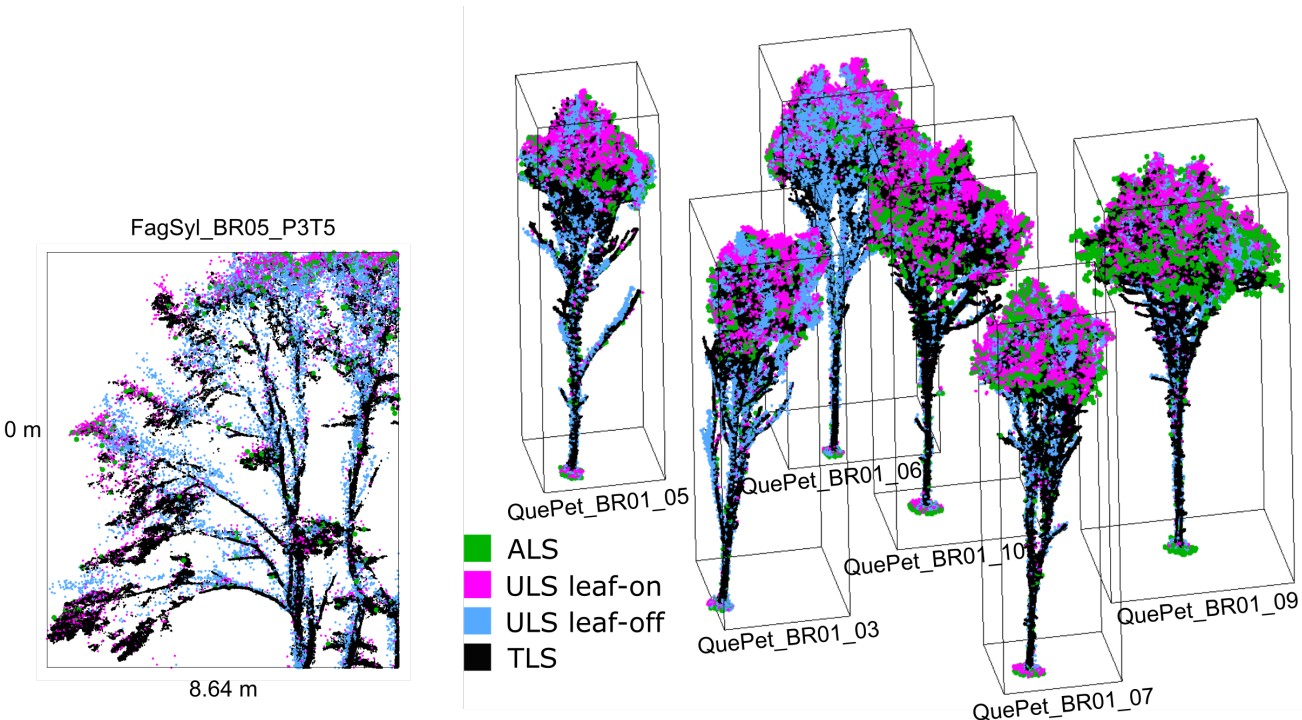

**Figure 4.** Point cloud visualizations showing the different point density and coverage for each platform. Left: Section of point clouds of a beech tree in plot BR05. The section has a depth of 1 m. Right: Point clouds of oaks in plot BR01. The trees are 16 m to 18 m tall. Point clouds are labeled by their tree ID in the dataset.

## 2.3 Tree point cloud extraction

We used different procedures to extract single tree point clouds from the laser scanning data (Figure 5). The full TLS point clouds were automatically segmented into individual tree point clouds (section 2.3.1). For the trees of interest in the center of the acquired scene, the segmentation was improved by manual editing. ULS and ALS tree point clouds were delineated manually in the point clouds. Because our multi-platform laser scanning point clouds are georeferenced and spatially overlapping, it was possible to extract trees from one point cloud using already extracted tree point clouds from another platform or acquisition.

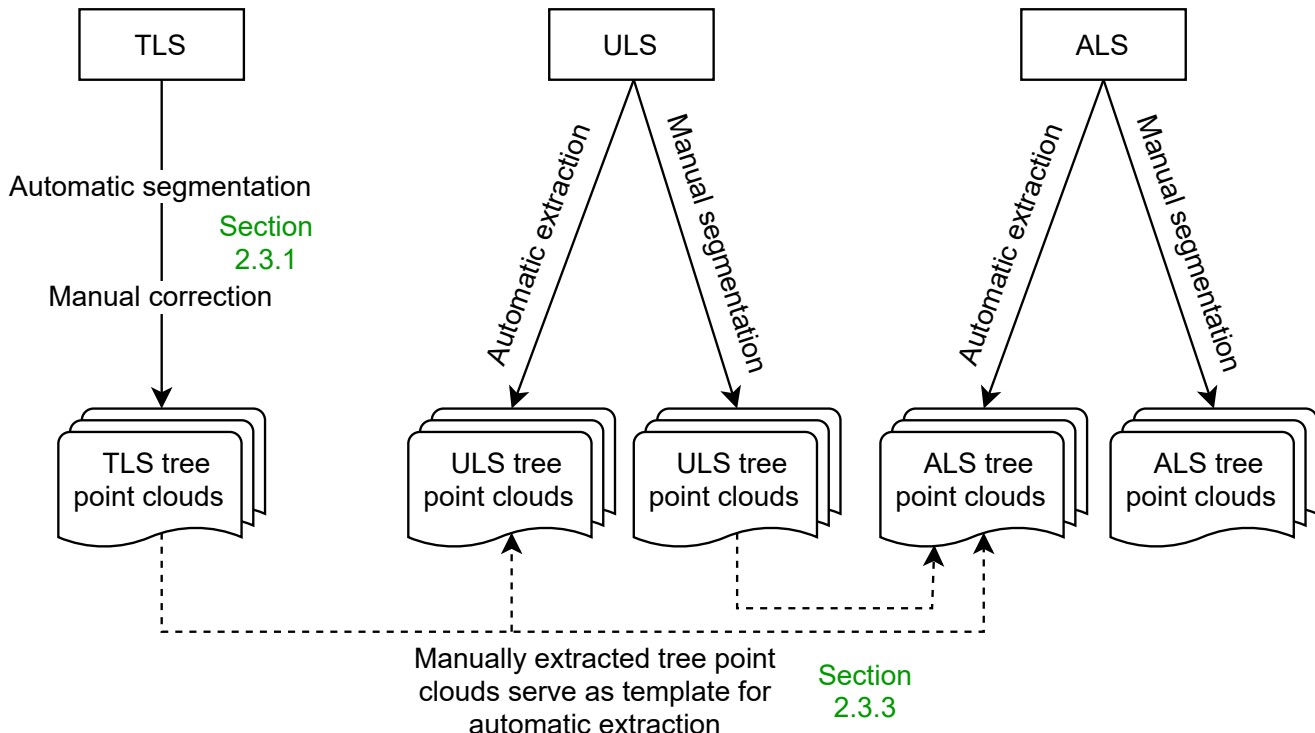

**Figure 5.** The different ways of single tree extraction. TLS point clouds were extracted with a combination of automatic segmentation and manual correction. ULS and ALS point clouds were partly segmented manually and partly extracted using a higher resolution point cloud from another platform (i.e., TLS and ULS).

This way, we only had to delineate each tree in one of the overlapping point clouds. Table 5 and Figure 8 give an overview of the number of tree point clouds that are available for each data source, i.e., for each platform and canopy condition.

### 2.3.1 Automatic tree segmentation from TLS point clouds and manual post-processing

264 trees were extracted from the TLS point clouds. We first automatically segmented each TLS plot and then refined the segmentation result for our selected trees by manually editing the point clouds.

Automatic segmentation was performed with CompuTree software (Computree Group 2017, version 5.0.054b). The workflow consists of an Euclidean clustering approach combined with a competitive Dijkstra's algorithm and relies largely on the SimpleTree plugin (Hackenberg 2017, Beta version 4.33.06). After importing the point cloud of the forest scene, a digital terrain model (DTM) is computed. Next, a slice of the point cloud is extracted between $1.0\,\mathrm{m}$ and $1.6\,\mathrm{m}$ above the DTM. The sliced point cloud is spatially clustered to derive one seed cluster per tree stem. Each point in a stem seed is assigned a cluster ID and an initial distance of 0. A competitive Dijkstra's algorithm is run, where each vegetation point is connected to one of

the clusters and receives the according cluster ID. A graph is built from the connected points. If there are multiple possible paths between a seed point and a target point, the shortest path is chosen.

The result of this automatic approach is error-prone, i.e., the Dijkstra's algorithm cannot guarantee the assignment of each point to the correct tree cluster. This is due to overlapping tree crowns and occlusions in the crown. Furthermore, errors in the clustering of stems seeds can lead to false labeling of points.

We applied a manual correction procedure. Selected tree point clouds were compared to surrounding segmented trees to identify erroneous or missing points and correct them manually. This was done in two stages by independent editors using the

interactive segmentation tool of CloudCompare (CloudCompare, 2019). Human experts can take into account the morphology of the crowns in ways which are barely feasible automatically. A combined approach of automatic segmentation and manual correction is therefore a good solution to precisely delineate trees.

### 2.3.2 Manual tree extraction from ALS and ULS point clouds

From ALS and ULS point clouds, we delineated trees manually in the point cloud using the interactive segmentation tool of

CloudCompare or we used already manually delineated trees as templates for automatic extraction (section 2.3.3).

### 2.3.3 Automatic tree extraction from ALS and ULS point clouds using extracted template tree point clouds

Because our laser scanning point clouds are spatially overlapping and co-registered, we were able to use already extracted tree point clouds from one dataset to query points from another dataset. For each point in the source tree point cloud, we queried all nearest neighbors within a given search radius in the larger area target point cloud to extract the same tree in the target point

cloud. This was done with a k-nearest neighbor (kNN) search using a kDTree. We ensured that the source point clouds were of equal or higher resolution than the target point clouds to avoid extrapolation. For instance, we extracted ULS point clouds using TLS source point clouds but not ALS source point clouds because this could lead to missing points if, e.g., the stem was not completely captured in the ALS point cloud. When using TLS point clouds as source point clouds, we used search radii of $0.3\,\mathrm{m}$, $0.5\,\mathrm{m}$ and $1.0\,\mathrm{m}$ in the nearest neighbor search. We then visually examined the extracted ALS and ULS point clouds and

chose the search radius achieving the best trade-off between errors of omission and errors of commission. For the extraction using ULS template point clouds, we always used a search radius of $1.0\,\mathrm{m}$.

### 2.3.4 Tree positions

We define the tree position as the intersection of the tree stem center with the terrain. To derive the tree positions, we first computed digital terrain models (DTMs) from the ALS point clouds. We then cut a slice through the bottom of the tree point

cloud, preferably the TLS or ULS point clouds, if available. We determined the x-y position of the stem as the means of the x and y coordinates of all points in the slice. At this position, we queried the elevation of the DTM to retrieve the matching z-value. For quality control, the derived positions were visually investigated. In some cases, they had to be corrected either

because of DTM errors or because the point cloud did not have enough stem points. If not enough stem points were present to determine the tree stem center at ground level, the x-y positions of the lowest visible stem point was used.

## 2.4 Tree measurements

Tree measurements were recorded in the field and/or derived from the point clouds. Table 3 gives an overview of the metrics and the data source they were derived from.

**Table 3.** Tree measurements and the source(s) they were recorded or computed from. Tree height was only measured from TLS point clouds for understory trees, for which tree height measurements from ALS or ULS would not be reliable. Tree height was only recorded in FM for smaller trees, where the top of the crown was visible from the ground (section 2.4.1). ALS = Airborne laser scanning, ULS = UAV-borne laser scanning, TLS = Terrestrial laser scanning, FM = Field measurements. All height measurements refer to height above the ground level at the stem position. For each tree, there can be multiple values for one tree metric if they were derived from different sources.

| | Metric | Unit | Description | Source(s) | | | |
|---|---|---|---|---|---|---|---|
| 1 | Tree height | m | Height of the tree | ALS | ULS leaf-on | (TLS) | (FM) |
| 2 | Diameter at breast height (DBH) | cm | Stem diameter, measured at $1.3\,\mathrm{m}$ above ground | - | - | TLS | FM |
| 3 | Crown base height (CBH) | m | Height of the lowest branch | ALS | ULS | - | FM |
| 4 | Crown base height (CBH) green | m | Height of the lowest green (living) branch | - | - | - | FM |
| 5 | Mean crown diameter (CD) | m | Mean of two orthogonal crown diameters | ALS | ULS | - | FM |
| 6 | Crown projection area (CPA) | $\mathrm{m}^2$ | Area of the tree crown | ALS | ULS | - | - |

### 2.4.1 Field measurements

Field measurements were conducted in six plots of $1\,\mathrm{ha}$ (Figure 2): BR01, BR03 and BR05 in Bretten and KA09, KA10 and KA11 in Karlsruhe. For each tree, the diameter at breast height (DBH) was measured with a measuring tape at $1.3\,\mathrm{m}$ above ground. To determine the crown diameter (CD), the diameter of the estimated dripline (i.e., the line right under the outer circumference of the crown) was measured in two orthogonal directions using a Haglöf Vertex-IV hypsometer. Mean CD was then calculated as the mean of these two distance measures. We define crown base height (CBH) as the height of the lowest branch with a minimum length of $1\,\mathrm{m}$ and crown base height green (CBH green) as the height of the lowest green (living) branch with a minimum length of $1\,\mathrm{m}$. We selected the branches fulfilling these conditions by visual estimation and

then measured their heights above ground using the hypsometer. Since measuring tree height with the hypsometer is difficult in dense forest stands where high tree tops are barely visible, tree height was only measured for about $10\,\%$ of the trees. These were mostly understory trees, where the crown was in direct view. As additional attributes, tree species and state (alive or dead) were recorded. For trees outside of the six field measurement plots, only species and state were determined. Field measurements

and single tree point clouds were matched based on extracted tree positions, the distinct crown shape and branch architecture, and the local neighborhood of the trees. For this, additional field surveys were conducted, in which 3D views of the laser scanning point clouds were used to identify the corresponding trees in the field.

### 2.4.2 Tree metrics derived from point clouds

In addition to field measurements, tree metrics were estimated from the laser scanning point clouds. For the computation of the

230 metrics, the height values of each tree point cloud were height-normalized by subtracting the z-coordinate of the tree position (representing the terrain height at the location of the stem) from all z-coordinates in the point cloud.

DBH was estimated only from TLS tree point clouds. Due to the terrestrial view point and close range, many returns are recorded from the stems, allowing accurate DBH estimation. A slice of the tree point cloud was extracted at a height between $1.28\,\mathrm{m}$ and $1.32\,\mathrm{m}$ above ground. In case the stem was partly occluded, a thicker slice was extracted or the vertical position of

235 the slice was adapted to achieve better results. We projected the slice onto the x-y plane and fit a circle to the points using the random sample consensus (RANSAC) algorithm to derive the diameter using 1000 random samples. For leaning trees, errors may arise from the projection of points to the x-y plane instead of a plane perpendicular to the stem. However, since most of our TLS trees were upright, this error is negligible. Some tree stems have a rather elliptical than circular stem shape. In these cases (44 of 264 TLS point clouds), an ellipse was fitted to the projected points of the stem slice using least squares[1]. The DBH

was then determined as the arithmetic mean of the major and the minor axes of the ellipse. For all trees, the quality of the fit was controlled visually. All figures of the circle/ellipse fitting are included in the Supplement together with a list of trees for which a) a different slice height was used, b) the visual assessment suggests low accuracy and c) TLS-derived DBH values deviate more than $10\,\%$ from the values measured in the field.

Tree height was derived as the z-coordinate of the highest tree point, normalized by the elevation of the tree position, i.e.,

the base of the tree.

CBH was determined by dividing the point cloud into height sections of $0.1\,\mathrm{m}$ and calculating the maximum x-y distance between any two points in the section. Starting from the bottom, we iterate over the height sections. Once the maximum horizontal point distance exceeded a threshold of DBH $+\,1\,\mathrm{m}$ (if no DBH measurement was available, a default of $0.5\,\mathrm{m}$ was used), the center of the height section is defined as the CBH (Figure 6) and the iteration is terminated. Because of the fixed

section height, CBH estimates have decimeter resolution.

The tree crown is defined by all points above the CBH. Crown projection area (CPA) was calculated in two different ways:

a) as the area of the 2D (x-y) *convex hull* of all crown points using the Quickhull algorithm (Barber et al., 1996).

---

[1]as implemented by Nicky van Foreest: https://github.com/ndvanforeest/fit_ellipse

b) as the area of the 2D (x-y) *concave hull* of all crown points using the k-nearest neighbors approach by Moreira and
   Yasmina Santos (2007) as implemented by Craig (2017).

Lastly, mean CD was computed as the mean of the largest diameter and the largest perpendicular diameter of the concave
hull (Figure 6).

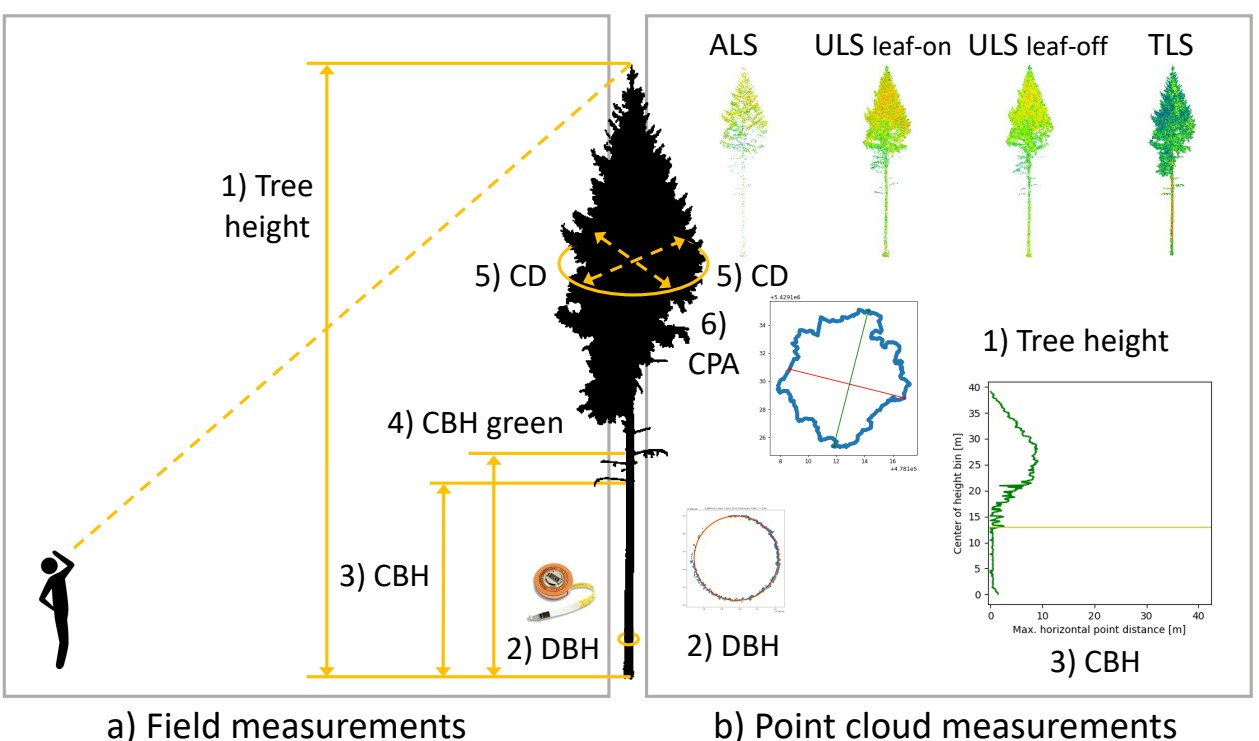

**Figure 6.** Overview of single tree measurements a) recorded in the field and b) derived from the point clouds. Numbers are corresponding to Table 3. Icon on the left ("man looking up") by DonBLC, https://thenounproject.com/. Picture of diameter tape from https://www.forestry-suppliers.com/.

## 3   Results

The data records are split into two parts:

1) The full TLS and ULS point clouds and extracted ALS subsets covering the ULS, TLS and field measurement plots (see
Figure 2)

2) The extracted single tree point clouds and respective measured and point cloud-derived tree metrics

The folder structure is visualized in Figure 7.

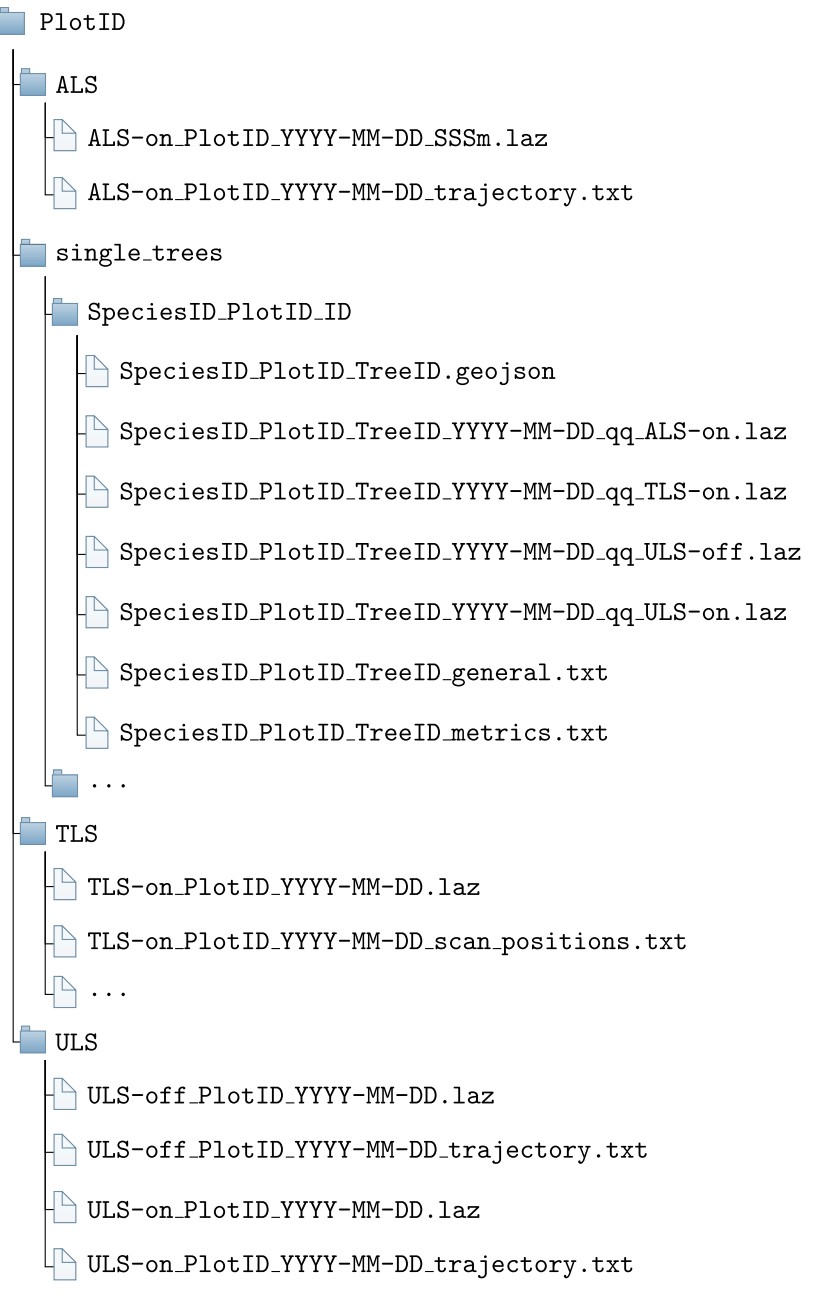

**Figure 7.** Folder structure for a single forest plot. PlotID = ID of the forest plot, e.g., BR01; SpeciesID = ID of the tree species, e.g., AcePse, see Table 4; TreeID = unique (in combination with PlotID and SpeciesID) identifier for the tree; YYYY-MM-DD = day of acquisition; qq = quality indicator, e.g., q2; on/off = canopy condition; SSSm = side length of the ALS forest plot in m, e.g. 300m. The ALS folder contains only one point cloud per plot. Multiple point clouds may be within the ULS folder due to different acquisition times and in the TLS folder due to different acquisition times and locations (in case of several acquisitions in the same plot on the same day, a suffix is added to distinguish the acquisitions, i.e., "_A", "_B", etc.). The single_trees folder always contains several subfolders, one for each tree.

### 3.1 LAS point format

All point clouds are stored in ASPRS LAS 1.4 format (ASPRS, 2013) or ASPRS LAS 1.2 (ASPRS, 2008), compressed (lossles) to LAZ. The field "Point source ID" holds the IDs of the flight lines or scan positions that the points originated from. All LAZ files contain the extra bytes amplitude [dB], reflectance [dB] and (pulse shape) deviation [] as exported by *RIEGL* systems. The amplitude is calibrated as the ratio of the optical power of the reflected light to the detection threshold of the instrument and depends on the distance of the target to the scanner (Pfennigbauer and Ullrich, 2010). The reflectance, however, is distance independent and determined by the properties of the target that was hit by the ray. Reflectance is given as the ratio of the amplitude of the hit target to the amplitude of the target the instrument was calibrated with, i.e., a diffuse white target (Pfennigbauer and Ullrich, 2010). The pulse shape deviation indicates the degree to which the shape of the received pulse deviates from a device-specific reference pulse shape.

### 3.2 Full forest plot or acquisition records

In order to also capture the neighborhood context of the single trees, understory and terrain information, we provide the larger-area georeferenced point clouds of each platform. These include 31 TLS point clouds stored in LAZ files. For each acquisition, we provide a tab-separated txt-file containing the coordinates of the scan positions. 20 LAZ-files are available containing the ULS point clouds acquired under leaf-on and leaf-off conditions as summarized in Table 2. Note that the ULS leaf-on data for plots BR03 and BR08 is included in one single LAZ file and no ULS data is available for plot BR04. For each acquisition, the trajectory is provided as a tab-separated txt-file, containing continuous records of the location and orientation of the sensor in the air. The "Time [s]" in the trajectory files corresponds with the "GPSTime" in the respective LAZ point cloud files and represents seconds since beginning of the respective GPS week (ASPRS, 2013). Finally, we provide 12 LAZ files with extracted ALS point clouds, one for each plot (Figure 2). ALS point clouds of plots BR01, BR03, BR05, KA09, KA10 and KA11 are $300 \, \mathrm{m} \times 300 \, \mathrm{m}$ in size and encompass the $100 \, \mathrm{m} \times 100 \, \mathrm{m}$ plots in which field measurements were recorded. ALS point clouds of plots BR02 and BR06 are $200 \, \mathrm{m} \times 200 \, \mathrm{m}$ and ALS point clouds of BR04, BR07, BR08 and SP02 are $140 \, \mathrm{m} \times 140 \, \mathrm{m}$ in size. For these six plots (BR02, BR04, BR06, BR07, BR08 and SP92), no field measurements are available. We provide the trajectory points for the respective parts of the flight strips covering the area in tab-separated txt-files. All ALS point clouds spatially overlap with the TLS and ULS acquisitions conducted in the respective plots.

All point clouds and the coordinates of the scan positions and trajectories are in the same coordinate reference system: ETRS89/UTM zone 32N; EPSG:25832; ellipsoidal height, GRS80.

### 3.3 Single tree point clouds and metrics

Point clouds and point cloud-derived metrics are provided for 1491 individual trees of in total 22 different species. An overview of tree species and the respective number of trees is given in Table 4. Field measured metrics are available for 1060 of the trees. Tree point clouds acquired from all three platforms, i.e., ALS, TLS, and ULS (leaf-on), are provided for 249 trees. 1168 tree point clouds were extracted both from ULS leaf-on and ULS leaf-off data. A summary of all available data types per tree

**Table 4.** Number of trees of each species in the dataset.

| Species | SpeciesID | Number of trees |
|---|---|---|
| *Abies alba* Mill. | AbiAlb | 25 |
| *Acer campestre* L. | AceCam | 7 |
| *Acer pseudoplatanus* L. | AcePse | 39 |
| *Betula pendula* Roth | BetPen | 6 |
| *Carpinus betulus* L. | CarBet | 94 |
| *Fagus sylvatica* L. | FagSyl | 398 |
| *Fraxinus excelsior* L. | FraExc | 11 |
| *Juglans regia* L. | JugReg | 19 |
| *Larix decidua* Mill. | LarDec | 30 |
| *Picea abies* (L.) H. Karst. | PicAbi | 205 |
| *Pinus sylvestris* L. | PinSyl | 158 |
| *Prunus avium* (L.) L. | PruAvi | 19 |
| *Prunus serotina* Ehrh., nom. cons. | PruSer | 7 |
| *Pseudotsuga menziesii* (Mirb.) Franco | PseMen | 191 |
| *Quercus petraea* (Matt.) Liebl. | QuePet | 156 |
| *Quercus robur* L. | QueRob | 7 |
| *Quercus rubra* L. | QueRub | 111 |
| *Robinia pseudoacacia* L. | RobPse | 1 |
| *Salix caprea* L. | SalCap | 1 |
| *Sorbus torminalis* (L.) Crantz | SorTor | 1 |
| *Tilia spec.* L. | TilSpec | 4 |
| *Tsuga heterophylla* (Raf.) Sarg. | TsuHet | 1 |
| | | 1491 |

is given in Table 5 and Figure 8. Figure 4 shows the different characteristics of the point clouds acquired from the different platforms and in the different canopy conditions. When comparing the TLS leaf-on point clouds with the ULS leaf-off point clouds, it is clearly visible that the branches hang down more in the summer when they are full of leaves than in the fall or winter when they are free of foliage.

There are multiple reasons for the different available data types per tree:

– TLS acquisitions did not cover the entire ALS/ULS acquisitions but only selected locations, so only 246 TLS point clouds of trees are available.

– No ULS acquisition was performed in plot BR04.

**Table 5.** Number of trees per plot and data source. Note that most trees were measured from different platforms and at different times. ULS leaf-off data is available for 1173 trees. Of these, 133 trees were measured in autumn 2019, 537 trees were measured in spring 2020, and 503 trees were measured both in autumn 2019 and spring 2020.

| Plot | Data source | | | | | | |
|------|-----|------------|------------------------------|-----------------------------|-----|--------------------|-----|
| | ALS | ULS (leaf-on) | ULS (leaf-off) Autumn 2019 | ULS (leaf-off) Spring 2020 | TLS | Field measurements | Any |
| BR01 | 514 | 503 | 503 | 503 | 15 | 410 | 514 |
| BR02 | 42 | 42 | - | 41 | 42 | - | 42 |
| BR03 | 195 | 141 | - | 141 | 41 | 162 | 200 |
| BR04 | 9 | - | - | - | 9 | - | 9 |
| BR05 | 278 | 278 | - | 278 | 35 | 224 | 278 |
| BR06 | 29 | 29 | - | 29 | 29 | - | 29 |
| BR07 | 15 | 16 | - | 15 | 16 | - | 16 |
| BR08 | 13 | 13 | - | 12 | 13 | - | 13 |
| SP02 | 17 | 17 | - | 21 | 22 | - | 22 |
| KA09 | 177 | 136 | 133 | - | 15 | 98 | 177 |
| KA10 | 40 | 14 | - | - | 11 | 35 | 40 |
| KA11 | 151 | 97 | - | - | 16 | 131 | 151 |
| | | | 636 | 1040 | | | |
| All | 1480 | 1286 | 1173 | | 264 | 1060 | 1491 |

– ALS point clouds of understory trees with very few points were discarded. This concerned only silver firs in plot SP02 (AbiAlb_SP02_01 to AbiAlb_SP02_05) which had between 11 and 247 ALS points.

– Some trees were only extracted manually from ALS point clouds and these were not used for automatic extraction due to the lower resolution compared to ULS and TLS point clouds, which would cause information loss (e.g., missing parts of the stem).

Tree properties are provided in two tab-separated txt-files:

– General tree characteristics such as the species and the tree position in geographic coordinates and in UTM coordinates are listed in a tab-separated txt-file with suffix "_general".

– Tree metrics are provided in a tab-separated txt-file with suffix "_metrics". There is one row for each measurement source (e.g., field inventory measurement (FI), ALS, etc.) and one column for each tree metric.

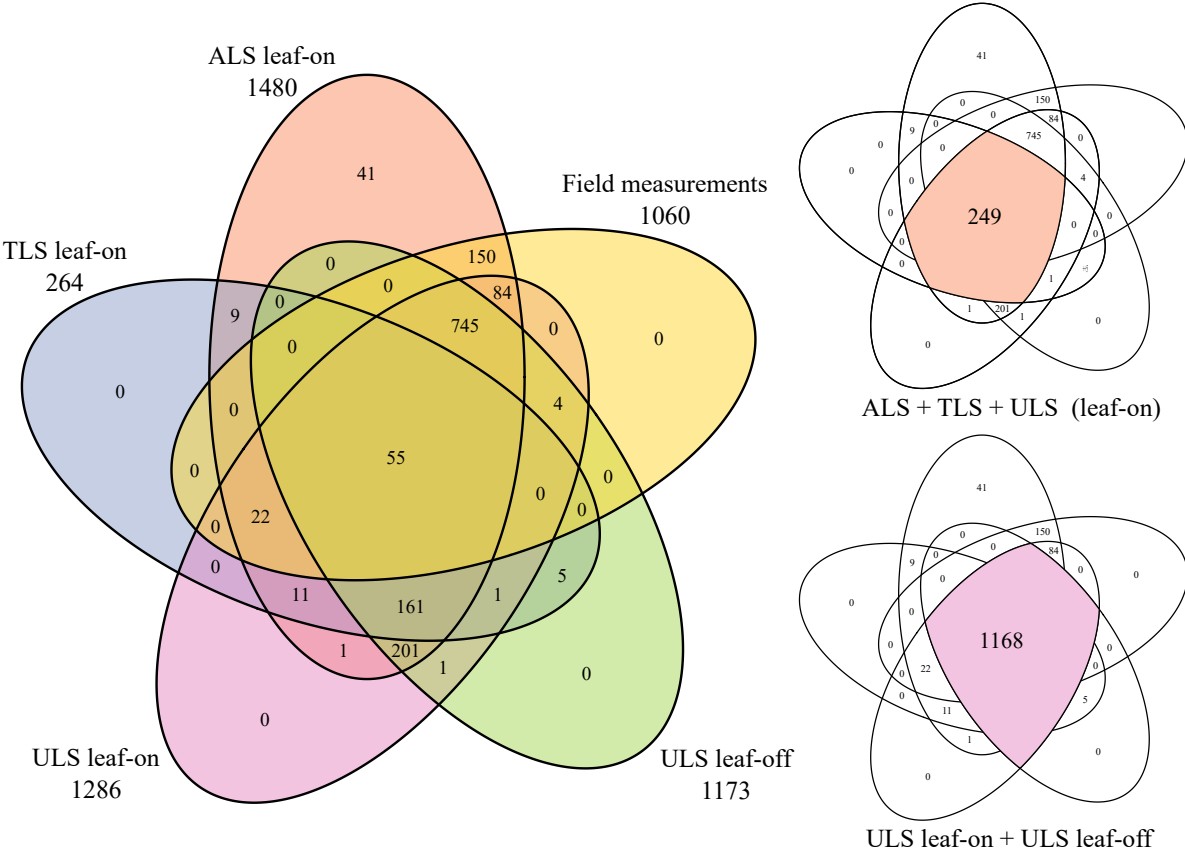

**Figure 8.** Venn diagrams showing the number of trees covered by the different acquisitions.

We furthermore provide all tree metadata in GeoJSON files, including geometry (position), properties (field measurements and point cloud metrics) and data (point cloud data filenames and metadata). These files can be loaded into common GIS software to visualize the tree positions and access the data.

## 4 Quality assessment

In the following, we describe how we assessed different aspects of data quality for our dataset. Table 6 summarizes all quality indicators related to point cloud alignment and georeferencing.

## 4.1 Airborne and UAV-borne laser scanning (ALS and ULS)

### 4.1.1 Positional accuracy

Positional accuracy of the ULS data was determined using roof-shaped wooden targets of $1\,\mathrm{m}$ length and $0.6\,\mathrm{m}$ width (for one side of the roof). The two planes of the roofs were recorded with four points each using a Leica Viva GS10/GS15 RTK GNSS unit tied into the SAPOS correction network (https://www.sapos-bw.de). A visual inspection of the wooden target point clouds and the GNSS reference measurements showed good alignment, with differences in the range that was expected due to the

limited scanner and RTK accuracies (Figure 9).

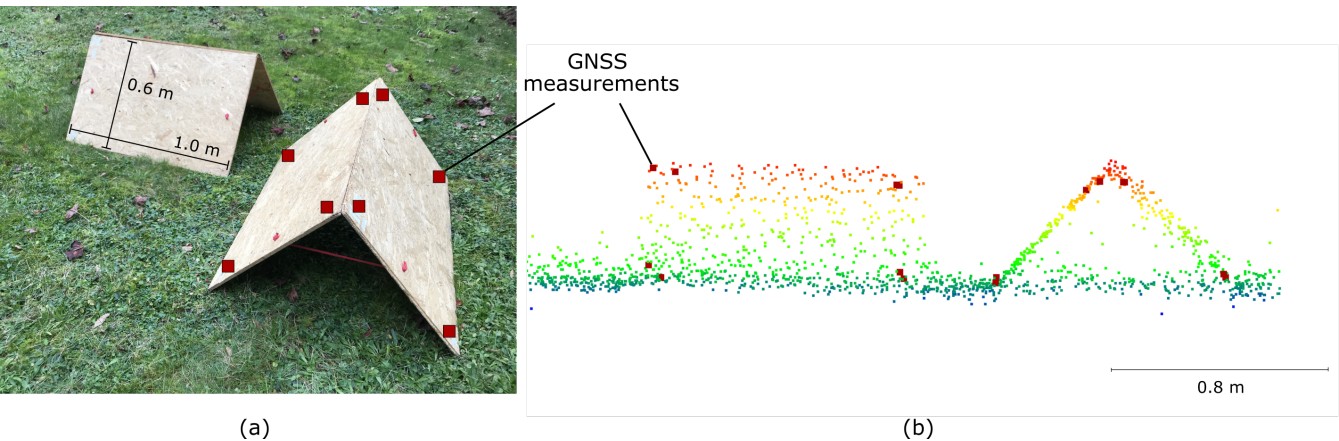

(a)  (b)

**Figure 9.** Roof-shaped wooden targets used for assessing the positional accuracy of the ULS point cloud. (a) Set-up in the field. The two planes of each roof are $1\,\mathrm{m}$ by $0.6\,\mathrm{m}$ in size. The red squares in both images mark the points, where GNSS measurements are taken. (b) View in a point cloud viewer. The ULS point cloud is colored by z-coordinate, the GNSS measurements are shown as large red squares. The georeferencing quality is considered good if the ULS points on the roofs agree closely with the GNSS reference measurements, as shown in this example.

Absolute altitude accuracy of ALS data was determined by MILAN Geoservices GmbH using reference tiles recorded with ULS by the 3DGeo Research Group of Heidelberg University. The height of selected reference points was compared to the median of ALS points within a radius of $0.2\,\mathrm{m}$ (Bretten) and $0.3\,\mathrm{m}$ (Karlsruhe). The mean altitude differences (standard deviations) were $-0.003\,\mathrm{m}$ ($\pm0.023\,\mathrm{m}$), $0.006\,\mathrm{m}$ ($\pm0.016\,\mathrm{m}$), $-0.016\,\mathrm{m}$ ($\pm0.026\,\mathrm{m}$), $-0.028\,\mathrm{m}$ and ($\pm0.029\,\mathrm{m}$) for the four

tiles in Bretten and $-0.016\,\mathrm{m}$ ($\pm0.017\,\mathrm{m}$) for the tile in Karlsruhe. The quality of these reference tiles was, in turn, ensured by the same method as for the ULS data in general, see above.

Positional (horizontal) accuracy of the ALS data was controlled by MILAN Geoservice GmbH using building edges (i.e., stable surfaces). A building layout derived from TLS data, which was acquired by the 3DGeo Research Group and georefenced with RTK GNSS, was overlaid with the building layout derived by digitizing the classified ALS point cloud to control the

positional accuracy.

A further independent assessment of positional and altitude accuracy was conducted by the 3DGeo Research Group Heidelberg by comparing points sampled by ULS and ALS on roofs and other stable surfaces in control areas. Figures of the visual assessment are included in Appendix B in the metadata document published along with the dataset on the data repository PANGAEA (https://download.pangaea.de/reference/109167/attachments/SYSSIFOSS_2019-2020_meta_4.pdf, Weiser et al.
2022a). To obtain quantitative measures, we computed cloud-to-cloud distances between these ALS and ULS point clouds of the stable areas (six in Karlsruhe, three in Bretten). Resulting mean distances were between $0.03\,\mathrm{m}$ and $0.06\,\mathrm{m}$ with standard deviations between $0.01\,\mathrm{m}$ and $0.023\,\mathrm{m}$. This confirmed the high accuracy of the alignment of the datasets.

### 4.1.2  Strip alignment

The internal alignment of ALS and ULS flight strips was investigated by the 3DGeo Research Group by quantifying the
height differences between the flight strips with OPALS (version 2.3.2). For ALS, strip differences were quantified for the extracted plot point clouds (with the extents as shown in Figure 2). The ULS point clouds were cropped to the 1 ha plots for the assessment of the strip alignment to exclude points at the flight line edges.

For both datasets, single strip point clouds were first filtered for last returns. Second, grids of the standard deviation of the interpolated height (sigmaZ) and of the distances between grid points and the center of gravity of data points (excentricity)
were computed to derive a mask of areas suited for the quality check. For the quantification of strip differences, areas with sigmaZ $\geq 0.1\,\mathrm{m}$ and excentricity $\geq 0.8\,\mathrm{m}$ were masked. This concerns most areas with high vegetation, where strip differences cannot be quantified reliably.

For the ALS point clouds, median absolute strip differences were below $5\,\mathrm{mm}$ for 10 of 12 plots. The maximum median value was $11\,\mathrm{mm}$. $\sigma(\mathrm{MAD})$ was between $17\,\mathrm{mm}$ and $51\,\mathrm{mm}$[2]. For ULS leaf-on point clouds, median absolute differences were
up to $14\,\mathrm{mm}$ (mean: up to $55\,\mathrm{mm}$). For ULS leaf-off point clouds, median absolute strip differences were only up to $2\,\mathrm{mm}$ (mean: below $20\,\mathrm{mm}$ for 10 acquisitions, $51\,\mathrm{mm}$ and $62\,\mathrm{mm}$ for the remaining two acquisitions), suggesting that the large values in the leaf-on differences are remaining influence from vegetation. Histograms and maps of strip differences for ALS and ULS data are included in Appendix A in the metadata document published along with the dataset on the data repository PANGAEA (Weiser et al., 2022a).

## 4.2  Terrestrial laser scanning (TLS)

### 4.2.1  Positional accuracy and alignment to ULS point clouds

TLS point clouds were georeferenced by coarsely registering them to the ULS data via tree stem matching, followed by a fine alignment with the ICP-algorithm. Before running the ICP, we created downsampled TLS point clouds with the octree filter in RiSCAN Pro with voxel sizes of $0.05\,\mathrm{m}$. For each voxel of the octree, a new point is created by taking the center of gravity
for all the points within each voxel. We performed an ICP adjustment with five iterations, keeping the ULS point cloud fixed

---

[2]$\sigma(\mathrm{MAD})$ is the median absolute deviation to median (MAD) scaled by $1.4826$ to obtain a robust estimator for the standard deviation $\sigma$ under the assumption of a normal distribution, https://opals.geo.tuwien.ac.at/html/stable/ModuleHisto.html

while transforming the TLS point cloud. With these settings, we achieved mean point-to-plane distances of less than $2\,\text{mm}$ with standard deviations between $120\,\text{mm}$ and $140\,\text{mm}$.

For 13 of the 31 point clouds, we still observed significant shifts in a visual assessment and repeated the ICP with revised settings: We used the ULS point cloud as moving, preferably the one acquired under leaf-off canopy condition (if available) due to the better representation of the stems, and the TLS point clouds as fixed point clouds. This means, that the subset of points, for which correspondences are established and to which planes are fitted to subsequently minimize point-to-plane distances are selected from the TLS point cloud. This is likely to achieve better results, because the TLS point cloud has higher point density (allowing for better plane fit) and higher accuracy. We used the full resolution TLS point clouds and in some cases filtered for points below $15\,\text{m}$ or $10\,\text{m}$ to exclude crown points. For the tree crowns, the coverage of TLS and ULS point clouds is very different and there are many moving objects (branches and leaves). We increased the number of iterations to up to 10. We then applied the inverse output transformation matrix to the TLS point cloud to ensure that the final georeferencing is derived from the ULS data. This resulted in mean point-to-plane distances of below $4\,\text{mm}$ with standard deviations of $50\,\text{mm}$–$100\,\text{mm}$. More importantly, the visual alignment improved considerably.

Because of the lack of bigger planar areas in the scene, the influence of moving branches and leaves, the different viewing geometries and resulting occlusions of the ULS and TLS point clouds, and the sensor accuracy limits, it is expected that no sub-millimeter point-to-plane distances were achieved and the standard deviation is comparably high. The final alignment is sufficient for our main intention, i.e., the extraction of trees from ULS and ALS point clouds using template TLS point clouds. A visual impression of the quality of the alignment between the datasets acquired from the different platforms is given in Figure 4. The visual assessment and the low mean distances suggest no remaining systematic shifts.

### 4.2.2 Internal co-registration of single scan positions

Co-registration quality of scan positions relative to the main stable position for each plot was assessed in two stages. In the first stage we investigated differences computed in the least squares algorithm of the Multi Station Adjustment performed in RiSCAN Pro. The overall average of the standard deviation of registration error (RMSE) was $5\,\text{mm}$ with a maximum of $8\,\text{mm}$. In the second stage, a visual assessment was conducted by projecting a narrow section of the point cloud onto a plane and coloring the points by their scan position. We used stem slices for assessing horizontal registration errors and cylindrical reflective targets for assessing vertical registration errors. Horizontal stem slices were manually extracted for selected trees with an approximate section width of $0.2\,\text{m}$. To minimize the influence of wind effects, stem slices were cut out from low heights above ground where tree stems are very stable. This was especially relevant for younger trees where tree stems are smaller in diameter and therefore more prone to movement. No significant vertical registration errors were present in any of the plots. In most of the acquisitions (26 of 31), no or very small ($< 10\,\text{mm}$) horizontal registration errors were observed. In five acquisitions the maximum horizontal registration error was estimated to be between $10\,\text{mm}$ and $20\,\text{mm}$. Appendix C in the metadata document published along with the data (Weiser et al., 2022a) shows the assessment of horizontal and vertical TLS registration errors.

**Table 6.** Summary of the different indicators for georeferencing and alignment quality. SD = Standard deviation.

| Quality aspect | Quality indicator | Quantitative results | Qualitative results |
|---|---|---|---|
| ULS georeferencing accuracy | Agreement of GNSS measurements on roof-shaped targets with ULS points (section 4.1.1) | | Visual assessment showed no evident systematic errors |
| ALS georeferencing accuracy | Height difference between ALS and ULS point clouds in stable areas (section 4.1.1) | Mean (per area) = −0.028 m–0.006 m SD (per area) = 0.016 m–0.029 m | |
| | Cloud-to-cloud distances between ALS and ULS point clouds in (different) stable areas (section 4.1.1) | Mean (per area) = 0.03 m–0.06 m SD (per area) = 0.01 m–0.023 m | Visual assessment (Weiser et al. 2022a, Appendix B) |
| | Agreement of a TLS-derived building layout with the ALS point cloud | | Visual assessment showed no evident systematic errors |
| ULS strip alignment | Flight strip differences per acquisition (section 4.1.2) | Leaf-on: Median ≤ 0.014 m Leaf-off: Median ≤ 0.002 m | Histograms and maps in Weiser et al. 2022a, Appendix A |
| ALS strip alignment | Flight strip differences per plot (section 4.1.2) | Median ≤ 0.005 m for 10 of 12 plots, Largest median value = 0.011 m | Histograms and maps in Weiser et al. 2022a, Appendix A |
| TLS scan position alignment | RMSE of the registration error (as quantified in RiSCAN) | Mean (over all acquisitions): 0.005 m Largest mean value = 0.008 m | |
| TLS scan position alignment | Differences between point clouds from different scan positions, quantified on stem slices (section 4.2.2) | ≤ 0.02 m | Visual assessment (Weiser et al. 2022a, Appendix C) |
| TLS georeferencing | Point-to-plane distances between the TLS and ULS point clouds after the ICP adjustment (section 4.2.1) | Mean ≤ 0.004 m SD = 0.05 m–0.14 m | Visual assessment showed no evident systematic errors |

## 4.3 Quality of single tree point clouds

Tree point clouds were assigned a quality tag from highest (q1) to lowest (q6). Reasons for lower grades include

- segmentation errors, e.g., additional branches in the point cloud

- extraction errors, e.g., additional points from neighboring trees in the point cloud because of the fixed search radius used for the automatic kNN-based extraction

- missing parts/occlusion of tree parts

- wind effects (in case of TLS)

- scan position or flight strip alignment errors

The quality tags were subjectively assigned by different operators. For most trees that were extracted automatically using the kNN-algorithm, the quality was set to one lower than the source point cloud. The quality tag therefore provides a rough idea of the point cloud quality but is not guaranteed to be comparable across the entire dataset.

## 4.4 Quality of the tree metrics

The quality of the tree metrics computed from the tree point clouds depends on the accuracy of the tree point cloud segmentation and the point density of the tree point cloud. Error of commission in the segmentation may cause higher tree height, mean CD and CPA values and lower CBH values, while error of omission or a low point density may cause the opposite. Especially ALS tree point clouds of understory trees have a low point density and thus the computed metrics might differ from the field measured metrics. As field measurements were conducted from the ground, field measured height, mean CD and CBH of high trees have higher uncertainty than those of smaller trees. It is difficult to consider either the field measurements or the point cloud derived metrics as true reference. By comparing field measured and point cloud derived metrics, we can nevertheless get an idea of the error ranges and potential biases.

Tree metrics derived from different sources are highly correlated (except CBH). The Pearson correlation coefficient ($\rho$) of DBH measured in the field and DBH computed from the TLS tree point clouds is 0.98 and the root-mean-square error (RMSE) is only $3.5\,\mathrm{cm}$. The correlation of tree heights derived from different laser scanning point clouds is 1.00 and the RMSE is $0.3\,\mathrm{m}$–$0.4\,\mathrm{m}$. This is lower than the search radius of $0.5\,\mathrm{m}$ which was mostly used for the automatic tree extraction. The deviation of field measured tree height to point cloud tree height is higher (RMSE = $2.5\,\mathrm{m}$–$2.7\,\mathrm{m}$, $\rho$ = 0.96). Field measured tree heights might be less accurate especially in case of high trees with flat crowns, where the tree top was not visible from the ground. The RMSE of mean CD is $0.7\,\mathrm{m}$–$1.2\,\mathrm{m}$ between different point cloud metrics ($\rho$ = 0.97–0.98) and $1.5\,\mathrm{m}$–$1.6\,\mathrm{m}$ ($\rho$ = 0.88–0.90) between field measurements and point cloud metrics. Absolute crown diameters are between $2\,\mathrm{m}$ and $22.1\,\mathrm{m}$ with a median value of $7.7\,\mathrm{m}$ (values based on field measurements). Higher deviations in CD between field measurements and point cloud metrics occur for trees with an elongated horizontal crown extent where the direction of the two orthogonal diameter measurements is more important than for crowns with nearly circular circumference. The CD values computed from point

clouds might be overestimated for leaning trees, because the part of the stem which is not covered by the crown may still be included in the crown extent if it is above the estimated CBH. Field measured diameters might be of lower accuracy for high tree crowns where the outer crown circumference was difficult to estimate. The definition and determination of CBH is difficult both in the field and in the point cloud, resulting in high deviation of CBH values derived from different data sources. The RMSE of CBH computed from different laser scanning point clouds is $4.8\,\mathrm{m}$–$6.9\,\mathrm{m}$ ($\rho = 0.66$–$0.76$). On the one hand, the point-cloud derived CBH is sensitive to segmentation errors especially in case of the automatic tree extraction from template point clouds. If understory points are wrongly assigned to a tree point cloud, the CBH may be underestimated. On the other hand, the lowest branches may not be sampled well in ULS and especially ALS point clouds, hence the CBH may be overestimated. The field-measured CBH accuracy is negatively affected by an over- or underestimation of branch lengths resulting in a wrong selection of the lowest branch defining the CBH. The RMSE between field measured and point cloud derived CBH is $4.0\,\mathrm{m}$–$5.1\,\mathrm{m}$ and $\rho$ is $0.59$ –$0.72$ Figure 10 shows scatter plots of field measured compared to point cloud-derived DBH, tree height, mean CD and CBH.

## 5  Usage notes

There are different methods to clean point clouds prior to further processing, e.g., applying the statistical outlier removal (SOR, Rusu et al. 2008 ) as implemented, e.g., in the Point Cloud Library (PCL, Rusu and Cousins 2011) or in CloudCompare. When working with the TLS data, we recommend to filter the point cloud by pulse shape deviation (as defined by *RIEGL*). This deviation can give an idea of the reliability of the range measurement, so excluding points with high deviation can improve the overall quality of the point cloud (Pfennigbauer and Ullrich, 2010). We found a value of 50 to be a suitable threshold.

Because we define the tree position as the position of the stem at ground level, this position (in ETRS89 / UTM 32N coordinates; EPSG:25832, ellipsoidal height, GRS80) can be used to normalize the height values of the point clouds to derive heights above ground.

Point clouds can be visualized and processed using the open-source software CloudCompare, other LiDAR software like OPALS (Pfeifer et al., 2014) or LAStools (Rapidlasso GmbH, 2020), or custom code.

We created the Python software package `pytreedb` (https://github.com/3dgeo-heidelberg/pytreedb) which provides a simple database interface and REST API to perform queries on a database of tree objects. `pytreedb` allows selecting trees using different filter, viewing maps of the selection results and exporting data in different formats (LAZ point clouds, GeoJSON files, CSV files).

Software for the generation of tree models from TLS tree point clouds include TreeQSM (Raumonen, 2020; Raumonen et al., 2013), SimpleForest (Hackenberg et al., 2015; Hackenberg, 2021), PlantScan3D (Boudon, 2021) and AdTree (Du et al., 2019; Nan et al., 2021). These tools reconstruct the woody structure of the trees and therefore require input point clouds without leaf points. Leaves may be removed from TLS point clouds using spectral and geometric features, as proposed in multiple studies (Krishna Moorthy et al., 2020; Vicari et al., 2019; Wang et al., 2020; Yun et al., 2016; Zhou et al., 2019) and implemented in different software packages (TLSeparation, Vicari 2021; LeWoS, Wang 2020). Structural tree reconstruction algorithms may

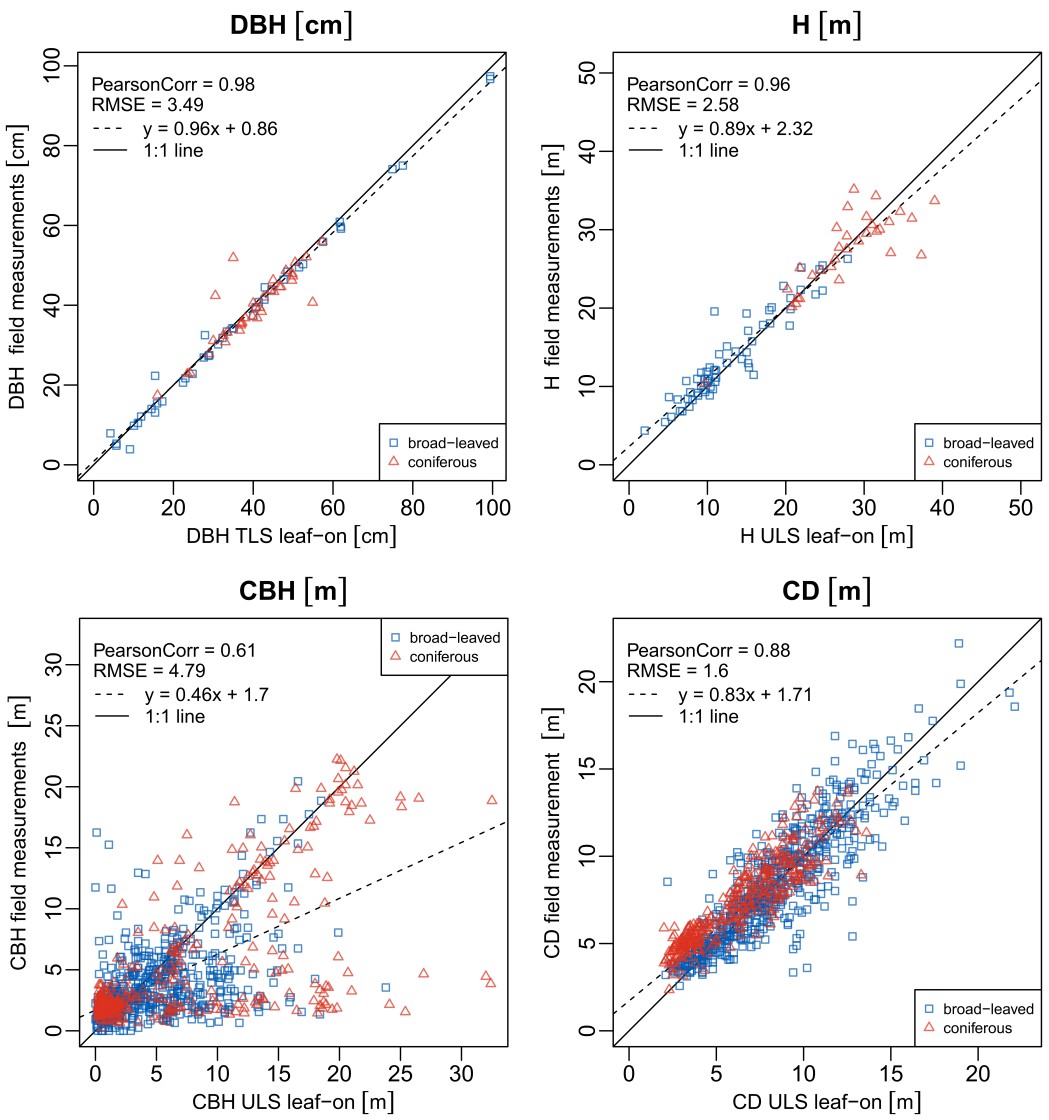

**Figure 10.** Scatter plots of metrics computed from tree point clouds and measured in the field. Pearson correlation, root-mean-square error and the equation of a linear model fit to the data are given. DBH = diameter at breast height, H = tree height, CBH = crown base height, CD = crown diameter.

also perform well on some of the ULS leaf-off point clouds, for which no leaf removal is required. Trees may furthermore be reconstructed using voxel-based approaches (Weiser et al., 2021).

To perform simulation studies with the reconstructed 3D forest scenes, e.g., to conduct sensitivity analyses, radiative transfer models such as DART (Gastellu-Etchegorry et al., 2004), librat (Calders et al., 2013; Lewis and Muller, 1993; Philip Lewis, 1999) or Rayspread (Widlowski et al., 2006) may be used. DART also allows to simulate LiDAR. Furthermore, there are specialized LiDAR simulators such as HELIOS++ (Winiwarter et al., 2022, 2021). The point clouds presented here can be used as scene parts for HELIOS++ after converting LAZ files to ASCII files, e.g. using LAStools:

```
las2txt -i <pointcloud.laz> -o <pointcloud.xyz> -oparse xyz
```

and using the xyzloader of HELIOS++ (https://github.com/3dgeo-heidelberg/helios/wiki/Scene#xyz-point-cloud-loader). Alternatively, the trees can be loaded as *DetailedVoxels* (https://github.com/3dgeo-heidelberg/helios/wiki/Scene#detailedvoxels), e.g., after deriving voxel-based plant area density estimates using the software AMAPVox (Vincent et al. 2017; Weiser et al. 2021, https://amap-dev.cirad.fr/projects/amapvox).

  Our dataset can furthermore be used as a benchmark and testing dataset for tree segmentation and species classification
algorithm. This was shown by Fu et al. (2022), who used the TLS point clouds of this dataset for testing the accuracy of their individual tree segmentation algorithm. It may be used to explore the differences between the point cloud measurements from different acquisition platforms and the tree and forestry metrics that can be derived from them. Our dataset can be valuable for developing and testing algorithms on the retrieval of forestry parameters on the single tree or the plot level. Furthermore, open forest datasets like this can serve as reference datasets for the calibration and validation of spaceborne data products
like biomass maps from satellite images or from spaceborne LiDAR data such as the NASA's Global Ecosystem Dynamics Investigation GEDI (Dubayah et al., 2020).

## 6 Conclusions

Our dataset encompasses terrestrial and airborne laser scanning point clouds acquired from three different platforms: a static tripod, an uncrewed aerial vehicle (UAV) and a fixed-wing aircraft. We thus obtain georeferenced point cloud representations
of the overlapping forest areas at different resolutions and from different viewpoints. UAV-borne laser scanning (ULS) point clouds were furthermore recorded in different season, under leaf-on and leaf-off canopy condition. From the full-coverage point clouds of the plot acquisitions, which are also provided within this publication, we identified and extracted individual single trees, resulting in 4205 point clouds of 1491 individual trees of 22 different species. Most of these trees were captured by airborne and UAV-borne laser scanning. For 249 trees, high-resolution terrestrial laser scanning point clouds are additionally
available. We recorded tree measurements in the field, which we compare to and complement with point cloud derived tree metrics. This results in single tree forest inventory data with one to five 3D point cloud representations of each tree. The full aerial point clouds, the single tree point clouds and the measured and point-cloud derived tree metrics have high reuse value for numerous applications. We identified the following main areas for such applications: i) Development of allometric equations

between 3D tree structures and forest inventory variables, ii) application as training data for machine learning applications such as tree instance segmentation, tree species classification or tree metric computation, and iii) creation of detailed tree models for LiDAR simulations or for computer rendering.

## 7 Code and data availability

Our dataset is published with PANGAEA and available under the DOI: https://doi.org/10.1594/PANGAEA.942856 (Weiser et al., 2022b). For each forest plot, one zip folder is provided, containing the forest plot point clouds with trajectory or scan position information and the single tree point clouds and measurements. Point clouds of the full-coverage airborne laser scanning acquisitions (covering around $65\,\mathrm{km^2}$ in Karlsruhe and $32\,\mathrm{km^2}$ in Bretten) and associated full-waveform data are not part of the PANGAEA dataset, but are available from the authors on reasonable request.

Python code used for processing the dataset is openly available on GitHub (https://github.com/3dgeo-heidelberg/syssifoss). The code uses the libraries NumPy, pandas, laspy, pykdtree, SciPy, jsonschema and Matplotlib. The scripts for georeferencing use the Python bindings of OPALS (Pfeifer et al., 2014) modules and were tested with versions 2.3.1 and 2.3.2. Please contact the authors for guidance on using the code.

## Appendix A

Table A1: Forest characteristics of full inventory plots of $1\,\mathrm{ha}$. For the number of trees and the basal area, both absolute (abs.) and relative (rel.) values are provided. It should be noted that point clouds were not extracted for every tree in these plots.

| Plot | Species | Number of trees | | Basal area [$\mathrm{m^2\,ha^{-1}}$] | | Percentage of dead trees | Median height [m] |
|---|---|---|---|---|---|---|---|
| | | abs. | rel. | abs. | rel. | | |
| **BR01** | **all** | **738** | **(100 %)** | **30.96** | **(100 %)** | **11 %** | **14.0** |
| | *Acer campestre* L. | 5 | (1 %) | 0.17 | (1 %) | 0 % | 7.9 |
| | *Acer pseudoplatanus* L. | 12 | (2 %) | 0.05 | (0 %) | 0 % | 9.8 |
| | *Carpinus betulus* L. | 118 | (16 %) | 1.10 | (4 %) | 1 % | 8.3 |
| | *Fagus sylvatica* L. | 225 | (30 %) | 15.29 | (49 %) | 1 % | 18.9 |
| | *Fraxinus excelsior* L. | 23 | (3 %) | 0.18 | (1 %) | 9 % | 12.0 |
| | *Larix decidua* Mill. | 6 | (1 %) | 0.90 | (3 %) | 0 % | 33.8 |
| | *Picea abies* (L.) H. Karst. | 163 | (22 %) | 3.86 | (12 %) | 34 % | 15.8 |
| | *Pinus sylvestris* L. | 2 | (0 %) | 0.43 | (1 %) | 0 % | 29.8 |
| | *Prunus avium* (L.) L. | 26 | (4 %) | 0.47 | (2 %) | 0 % | 13.8 |
| | | | | | | Continued on next page | |

| Plot | Species | Number of trees | | Basal area [m² ha⁻¹] | | Percentage of dead trees | Median height [m] |
|---|---|---|---|---|---|---|---|
| | | abs. | rel. | abs. | rel. | | |
| | *Pseudotsuga menziesii* (Mirb.) Franco | 21 | (3 %) | 5.66 | (18 %) | 0 % | 36.4 |
| | *Quercus petraea* (Matt.) Franco | 121 | (16 %) | 2.64 | (9 %) | 9 % | 13.7 |
| | *Salix caprea* L. | 1 | (0 %) | 0.05 | (0 %) | 0 % | 16.4 |
| | *Sambucus nigra* L. | 14 | (2 %) | 0.06 | (0 %) | 57 % | 5.8 |
| | unidentified coniferous | 1 | (0 %) | 0.10 | (0 %) | 100 % | 27.4 |
| **BR03** | **all** | **473** | **(100 %)** | **42.00** | **(100 %)** | **5 %** | **20.4** |
| | *Abies alba* Mill. | 28 | (6 %) | 3.75 | (9 %) | 4 % | 26.5 |
| | *Acer campestre* L. | 4 | (1 %) | 0.08 | (0 %) | 0 % | 10.4 |
| | *Acer pseudoplatanus* L. | 24 | (5 %) | 0.70 | (2 %) | 0 % | 9.2 |
| | *Betula pendula* Roth | 1 | (0 %) | 0.07 | (0 %) | 100 % | 18.2 |
| | *Carpinus betulus* L. | 123 | (26 %) | 1.82 | (4 %) | 5 % | 11.0 |
| | *Corylus avellana* L. | 3 | (1 %) | 0.01 | (0 %) | 33 % | 6.1 |
| | *Euonymus europaeus* L. | 2 | (0 %) | 0.00 | (0 %) | 0 % | 4.1 |
| | *Fagus sylvatica* L. | 50 | (11 %) | 2.21 | (5 %) | 0 % | 11.8 |
| | *Fraxinus excelsior* L. | 6 | (1 %) | 0.19 | (0 %) | 0 % | 20.8 |
| | *Juglans regia* L. | 2 | (0 %) | 0.00 | (0 %) | 0 % | 6.0 |
| | *Metasequoia glyptostroboides* Hu et Cheng | 2 | (0 %) | 0.16 | (0 %) | 0 % | 23.3 |
| | *Picea abies* (L.) H. Karst. | 10 | (2 %) | 0.51 | (1 %) | 20 % | 21.2 |
| | *Prunus avium* (L.) L. | 6 | (1 %) | 0.20 | (0 %) | 0 % | 13.1 |
| | *Pseudotsuga menziesii* (Mirb.) Franco | 164 | (35 %) | 28.04 | (67 %) | 2 % | 35.9 |
| | *Quercus robur* L. | 5 | (1 %) | 2.40 | (6 %) | 0 % | 27.8 |
| | *Sambucus nigra* L. | 8 | (2 %) | 0.02 | (0 %) | 0 % | 6.4 |
| | *Sorbus torminalis* (L.) Crantz | 1 | (0 %) | 0.01 | (0 %) | 0 % | 13.3 |
| | *Taxus baccata* L. | 18 | (4 %) | 0.89 | (2 %) | 33 % | 6.2 |
| | *Tilia spec.* L. | 9 | (2 %) | 0.46 | (1 %) | 11 % | 16.9 |
| | *Tsuga heterophylla* (Raf.) Sarg. | 3 | (1 %) | 0.43 | (1 %) | 0 % | 20.8 |
| | unidentified broad-leaved | 4 | (1 %) | 0.05 | (0 %) | 100 % | 13.8 |

| Plot | Species | Number of trees | | Basal area [m² ha⁻¹] | | Percentage of dead trees | Median height [m] |
|---|---|---|---|---|---|---|---|
| | | abs. | rel. | abs. | rel. | | |
| **BR05** | **all** | **331** | **(100 %)** | **33.28** | **(100 %)** | **5 %** | **26.6** |
| | *Betula pendula* Roth | 2 | (1 %) | 0.03 | (0 %) | 0 % | 11.4 |
| | *Carpinus betulus* L. | 17 | (5 %) | 0.96 | (3 %) | 0 % | 17.2 |
| | *Fagus sylvatica* L. | 148 | (45 %) | 12.15 | (37 %) | 5 % | 23.7 |
| | *Juglans regia* L. | 32 | (10 %) | 0.71 | (2 %) | 3 % | 15.8 |
| | *Larix decidua* Mill. | 21 | (6 %) | 4.10 | (12 %) | 0 % | 31.5 |
| | *Picea abies* (L.) H. Karst. | 42 | (13 %) | 3.45 | (10 %) | 5 % | 28.6 |
| | *Pinus sylvestris* L. | 6 | (2 %) | 1.56 | (5 %) | 33 % | 36.1 |
| | *Pseudotsuga menziesii* (Mirb.) Franco | 44 | (13 %) | 7.42 | (22 %) | 2 % | 37.7 |
| | *Quercus petraea* (Matt.) Franco | 17 | (5 %) | 2.86 | (9 %) | 0 % | 33.7 |
| | *Salix caprea* L. | 1 | (0 %) | 0.03 | (0 %) | 0 % | 18.4 |
| | unidentified coniferous | 1 | (0 %) | 0.01 | (0 %) | 100 % | 9.3 |
| **KA09** | **all** | **272** | **(100 %)** | **25.21** | **(100 %)** | **8 %** | **20.2** |
| | *Acer pseudoplatanus* L. | 1 | (0 %) | 0.01 | (0 %) | 0 % | 10.3 |
| | *Amelanchier lamarckii* (Raf.) F. G. Schroed. | 3 | (1 %) | 0.01 | (0 %) | 0 % | 10.0 |
| | *Betula pendula* Roth | 17 | (6 %) | 0.16 | (1 %) | 0 % | 11.8 |
| | *Carpinus betulus* L. | 17 | (6 %) | 0.08 | (0 %) | 0 % | 7.2 |
| | *Castanea sativa* Mill. | 2 | (1 %) | 0.01 | (0 %) | 0 % | 7.0 |
| | *Fagus sylvatica* L. | 101 | (37 %) | 7.68 | (30 %) | 0 % | 20.5 |
| | *Ilex aquifolium* L. | 5 | (2 %) | 0.02 | (0 %) | 0 % | 5.6 |
| | *Picea abies* (L.) H. Karst. | 32 | (12 %) | 3.20 | (13 %) | 50 % | 24.4 |
| | *Pinus sylvestris* L. | 64 | (24 %) | 13.32 | (53 %) | 2 % | 30.7 |
| | *Populus tremula* L. | 1 | (0 %) | 0.01 | (0 %) | 0 % | 10.8 |
| | *Prunus avium* (L.) L. | 5 | (2 %) | 0.02 | (0 %) | 20 % | 7.1 |
| | *Quercus rubra* L. | 13 | (5 %) | 0.65 | (3 %) | 0 % | 11.6 |
| | *Sambucus nigra* L. | 5 | (2 %) | 0.03 | (0 %) | 40 % | 4.4 |
| | *Sorbus aucuparia* L. | 3 | (1 %) | 0.01 | (0 %) | 0 % | 7.4 |

| Plot | Species | Number of trees | | Basal area [m² ha⁻¹] | | Percentage of dead trees | Median height [m] |
|---|---|---|---|---|---|---|---|
| | | abs. | rel. | abs. | rel. | | |
| | *Taxus baccata* L. | 1 | (0 %) | 0.00 | (0 %) | 0 % | 3.8 |
| | unidentified broad-leaved | 2 | (1 %) | 0.00 | (0 %) | 100 % | 6.3 |
| **KA10** | **all** | **426** | **(100 %)** | **23.06** | **(100 %)** | **3 %** | **11.7** |
| | *Acer pseudoplatanus* L. | 28 | (7 %) | 0.27 | (1 %) | 0 % | 10.1 |
| | *Aesculus hippocastanum* L. | 1 | (0 %) | 0.26 | (1 %) | 0 % | 17.3 |
| | *Betula pendula* Roth | 2 | (0 %) | 0.16 | (1 %) | 0 % | 19.7 |
| | *Carpinus betulus* L. | 108 | (25 %) | 3.82 | (17 %) | 4 % | 15.1 |
| | *Castanea sativa* Mill. | 11 | (3 %) | 0.15 | (1 %) | 0 % | 11.6 |
| | *Ilex aquifolium* L. | 7 | (2 %) | 0.04 | (0 %) | 0 % | 9.0 |
| | *Pinus sylvestris* L. | 27 | (6 %) | 5.02 | (22 %) | 7 % | 26.7 |
| | *Prunus serotina* Ehrh., nom. cons. | 140 | (33 %) | 0.88 | (4 %) | 1 % | 7.0 |
| | *Quercus rubra* L. | 91 | (21 %) | 12.30 | (53 %) | 0 % | 23.6 |
| | *Robinia pseudoacacia* L. | 3 | (1 %) | 0.07 | (0 %) | 0 % | 14.2 |
| | *Sambucus nigra* L. | 4 | (1 %) | 0.04 | (0 %) | 75 % | 6.2 |
| | *Tilia spec.* L. | 3 | (1 %) | 0.03 | (0 %) | 0 % | 8.9 |
| | unidentified broad-leaved | 1 | (0 %) | 0.02 | (0 %) | 100 % | 14.5 |
| **KA11** | **all** | **787** | **(100 %)** | **28.66** | **(100 %)** | **3 %** | **15.7** |
| | *Acer pseudoplatanus* L. | 4 | (1 %) | 0.02 | (0 %) | 0 % | 10.4 |
| | *Fagus sylvatica* L. | 309 | (39 %) | 3.65 | (13 %) | 4 % | 12.2 |
| | *Picea abies* (L.) H. Karst. | 10 | (1 %) | 0.20 | (1 %) | 50 % | 10.6 |
| | *Pinus sylvestris* L. | 98 | (12 %) | 10.62 | (37 %) | 1 % | 24.8 |
| | *Prunus serotina* Ehrh., nom. cons. | 118 | (15 %) | 1.26 | (4 %) | 4 % | 9.4 |
| | *Pseudotsuga menziesii* (Mirb.) Franco | 5 | (1 %) | 0.29 | (1 %) | 20 % | 29.3 |
| | *Quercus petraea* (Matt.) Liebl. | 18 | (2 %) | 0.71 | (2 %) | 0 % | 17.9 |
| | *Quercus rubra* L. | 219 | (28 %) | 11.82 | (41 %) | 0 % | 22.3 |
| | *Tilia spec.* L. | 5 | (1 %) | 0.09 | (0 %) | 0 % | 12.0 |
| | unidentified broad-leaved | 1 | (0 %) | 0.00 | (0 %) | 100 % | 9.0 |

*Author contributions.* B.H. and F.F. conceived the study, L.W., J.S., N.K. and H.W. collected and analyzed the data, F.F., J.S. and H.W. drafted the manuscript. All authors reviewed the manuscript.

*Competing interests.* The authors declare no competing interests.

*Acknowledgements.* This work was supported by the Deutsche Forschungsgemeinschaft (DFG, German Research Foundation) in the frame of the project SYSSIFOSS - 411263134 / 2019-2022. We acknowledge Christian Seitz, Marian Schimka, Katharina Anders, Paula Kuss, Veit Ulrich, Vivien Zahs, Felix Schiefer, Elham Shafeian, Denis Debroize, Annika Denner, Michael Ewald, Lioba Martin, Helen Hake, Tobias Steinert, Vanessa Rittlinger, Fabio Bothner, Louisa Lücking, Janina Schüssler, Angelo Mayer, Katharina Ruge, Johannes Brand, Lea Schraml, Naia Haltmeier, Carolin Klonner, Daria Baete, Thorben Schrempp, Lisa-Maricia Schwarz and Andressa Soarez Braz for their assistance with data acquisition and processing.

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
