# Peer review of "Individual tree point clouds and tree measurements from multi-platform laser scanning in German forests"

_Earth System Science Data, 2022_

## Referee Comment (RC1)

***REVIEW: Individual tree point clouds and tree measurements from multi-platform laser scanning in German forests***

**General comments:**

In this data description manuscript the authors present a dataset of spatially overlapping and georeferenced ALS, ULS and TLS tree point clouds (and their corresponding tree metrics) for 12 plots spread over 2 German mixed forests.

In general, the manuscript includes all the necessary information on how the dataset was acquired and how it can be used. This type of open access dataset containing co-registered point clouds from three different platforms (TLS, ULS and ALS) is new and certainly contributes to the research community. As the authors mention, the data can be used for a range of purposes including calibration and validation. The methods and materials are described in sufficient detail and are well written.

But the writing of the abstract and introduction should be improved. In the specific comments you can find some comments regarding the abstract. Regarding the introduction, it seems to focus on 3D modelling (from line 13 till line 38) which doesn't seem the right focus in my opinion. You don't provide 3D models in your dataset, and this is only one aspect the dataset could be used for. If the data is to be used by ecologists in general (not just LiDAR users), the introduction should introduce ALS, ULS and TLS (discuss their different perspectives) and what their spatially overlapping data and individual tree point clouds can be used for and why open access datasets with co-registered ALS, ULS and TLS are important.

In summary, my suggestion is that this manuscript needs minor revisions before publication.

**Specific comments:**

Abstract:

- The first two sentences should be rewritten, for example:
    - Line 2 (p1) "Such point clouds allow us to e.g., retrieve …" → "These point clouds allow us to retrieve detailed information on the individual tree and forest structure."
- I would like to see more specific information on the number of plots and forests, for example:
    - "We conducted airborne laser scanning (ALS), UAV-borne laser scanning (ULS), and terrestrial laser scanning (TLS) in two German mixed forests with species typical for Central Europe. We provide spatially overlapping, georeferenced TLS, ULS and ALS point clouds for 12 hectare plots."
- Line 5-7 (p1) You mention the tree metrics are derived from the point clouds and measured in the field. Yet, only for half of the plots the metrics are measured in the field. This should be mentioned here.

Introduction:

- Line 41 (p2), this sentence suggest 1491 trees were extracted from all the scanning platforms (which is not the case). So maybe start with:
    - "249 trees were extracted from both ALS, ULS and TLS. Another 1036 trees were extracted from both ALS and ULS, and xxx only from ALS."

Methods:

- It would be nice to have a table with for each plot (like Table 3) the availability of the data sources (with for example also the number of trees scanned in each plot as this is not clear from the manuscript). This will help the user to choose the plots their interested in because of the availability of certain data sources and number of trees.

| Plot | ALS | ULS (leaf-on) | ULS (leaf-off) | TLS | Field measurements |
|------|-----|---------------|----------------|-----|--------------------|
| BR01 | 300 | 250 | 250 | 10 | 10 |
| BR02 |  |  |  |  | - |
| BR03 |  |  |  |  |  |
| BR04 |  | - | - |  | - |
| BR05 |  |  |  |  |  |
| BR06 |  |  |  |  | - |
| BR07 |  |  |  |  | - |
| BR08 |  |  |  |  | - |
| SP02 |  |  |  |  | - |
| KA09 |  |  |  |  |  |
| KA10 |  |  | - |  |  |
| KA11 |  |  | - |  |  |

Data:

- It would be really nice for the users if there were a file available with a table with for each tree information on the data which is collected for it. In this way the users could write a quick script to determine which trees they wants to use (e.g. only trees where of a certain species, of which there is quality 2 ALS, ULS, TLS and field data).

| tree | species | Plot | ALS q1 | … | ALS q6 | ULS (leaf-on) q1 | ULS (leaf-off) q1 | TLS Q1 | Field measurements |
|------|---------|------|--------|---|--------|------------------|-------------------|--------|--------------------|
| P17T34 | PicAbi | BR01 | FALSE | … | TRUE | FALSE | FALSE | FALSE | FALSE |

**Technical corrections:**

Line 7 (p1) "Our dataset may be used for the creation of 3D tree models for radiative transfer modelling or LiDAR simulation studies or to fit allometric equations between point cloud metrics and forest inventory variables" → "… creation of 3D tree models for radiative transfer modelling, LiDAR simulation studies or to fit allometric equations between point cloud metrics and forest inventory variables."

Line 301 (p19) "A visual inspection of the the wooden target …". Remove one the.

Line 306 (p19) "… and 0.3 m(Karlsruhe)." Add a comma between 0.3 m and (Karlsruhe).

Line 347 (p21) "… point clouds as fix point clouds." Shouldn't fix be fixed?

Line 431 (p29) " … and AdTree(Du et al., 2019…)." Add a comma between AdTree and (Du.

---

## Referee Comment (RC2)

**Individual tree point clouds and tree measurements from multi-platform laser scanning in German forests**. Weiser et al.

**General comments**

The paper outlines a large, interesting and well-structured data set derived from ALS, ULS and TLS point clouds acquired from twelve forest plots in southwestern Germany. These LiDAR derived data are supported by traditional forest measurements. The whole dataset is available for download representing a seldomly seem large scale open access dataset encompassing LiDAR data in central Europe.

The text is well written using a high standard of (American) English in the conventional scientific style, citations are both relevant and correctly presented.

**In my opinion this manuscript can be published with minor** revisions specific comments and suggestions of technical corrections are given below.

**Specific comments**

There is a lack of information about the specific forest stands in regards to their management history, composition and structure. This is important for the end user to know in order to make comparisons within and between datasets for comparability. Within this idea, some basic stand-based metrics for each of the sample plots might be of interest for the reader (species mixture, basel area etc.) such information can be included in a supplementary data table and will aid the end user in determining which data sets (study plots) to use, maybe in conjunction with their own data.

Figure 4 nicely illustrates the effect of leaf weight on branch position in the ULS data (blue vs. pink points) – this could be highlighted in a comment for added value.

Line 226/227 "For all trees, the quality of the fit was controlled visually." Please briefly expand on this, with what threshold, what was done if the ellipse was not fitting? Visual examination is very subjective, can consistency be assured? The users of the data must understand how this was carried out.

Line 280: was there a threshold used to discard trees? More detail needed.

**Technical corrections**

Splitting of words between lines – this might be a journal/template issue. While acceptable in German text But the splitting of words in English can be considered bad form. This is particularly apparent in line 152/153 across pages 9 and 10 overlap-ping.

Line 3/Line 34: Please also define UAV as "unmanned aerial vehicle" in the first instance, later defined in Line 456 as "uncrewed" this is better in a gender neutral usage.

Line 24: consider changing "The tree crowns' geometries…." to "The geometry of tree crowns can then…."

Line 42: Southwest → southwest, not a name of a place or region (please check the remainder of the document.

Line 54: consider changing to "the creation of 3D…."

Line 56: consider changing to "the development of…"

Line 222: slanted → leaning

Line 225: we fit an ellipse → an ellipse was fitted

Line 238: Concave → Convex

Line 320: change to read "the good alignment *of the data sets*."

Line 324: 1 -ha → 1 ha

Line 342: 5 iterations → five iterations (please check the rest of the document and change accordingly e.g. line 353).

Figure 10: attention to capitalisation of axis labels

Caption fig 10: overlap with page number

Line 466: like → Such as

---

## Author Response (AR1)

Dear colleagues and reviewers,

thank you for providing the reviews for our manuscript "Individual tree point clouds and tree measurements from multi-platform laser scanning in German forests". We made improvements to the manuscript according to your suggestions. Below, we provide our responses to the comments in *italics and blue* and excerpts from the updated manuscript in *italics and orange*.

In addition to the revised manuscript, we provide a version with highlighted changes as PDF file. Line numbers in the responses below refer to the version with tracked changes.

We have done a correction to one of the point clouds of our dataset and therefore provide a different DOI than in the original submission. Both the old DOI (https://doi.org/10.1594/PANGAEA.933426) and the new DOI (https://doi.org/10.1594/PANGAEA.942856; DOI registration in progress, currently available under https://doi.pangaea.de/10.1594/PANGAEA.942856) will – once the review period is finished – be freely accessible on PANGAEA. We furthermore corrected data acquisition settings (scan frequencies) reported in Table 2 and Table 3.

Lastly, we submit a new supplementary file (related to the circle/ellipse fitting for tree diameter at breast height estimation, based on the referee suggestions).

Yours sincerely,

Hannah Weiser

On behalf of all co-authors

**Referee comment 1**

**General comments:**

In this data description manuscript the authors present a dataset of spatially overlapping and georeferenced ALS, ULS and TLS tree point clouds (and their corresponding tree metrics) for 12 plots spread over 2 German mixed forests.

In general, the manuscript includes all the necessary information on how the dataset was acquired and how it can be used. This type of open access dataset containing co-registered point clouds from three different platforms (TLS, ULS and ALS) is new and certainly contributes to the research community. As the authors mention, the data can be used for a range of purposes including calibration and validation. The methods and materials are described in sufficient detail and are well written.

But the writing of the abstract and introduction should be improved. In the specific comments you can find some comments regarding the abstract. Regarding the introduction, it seems to focus on 3D modelling (from line 13 till line 38) which doesn't seem the right focus in my opinion. You don't provide 3D models in your dataset, and this is only one aspect the dataset could be used for. If the data is to be used by ecologists in general (not just LiDAR users), the introduction should introduce ALS, ULS and TLS (discuss their different perspectives) and what their spatially overlapping data and individual tree point clouds can be used for and why open access datasets with co-registered ALS, ULS and TLS are important.
*We revised the Introduction according to your suggestions (L 15-44).*

In summary, my suggestion is that this manuscript needs minor revisions before publication.

**Specific comments:**

Abstract:

- The first two sentences should be rewritten, for example: o Line 2 (p1) "Such point clouds allow us to e.g., retrieve …" → "These point clouds allow us to retrieve detailed information on the individual tree and forest structure."
- I would like to see more specific information on the number of plots and forests, for example:
  - "We conducted airborne laser scanning (ALS), UAV-borne laser scanning (ULS), and terrestrial laser scanning (TLS) in two German mixed forests with species typical for Central Europe. We provide spatially overlapping, georeferenced TLS, ULS and ALS point clouds for 12 hectare plots."

*We included your suggestions in the revised abstract (L 2-6).*
*These point clouds allow us to retrieve detailed information on the individual tree and forest structure. We conducted airborne laser scanning (ALS), uncrewed aerial vehicle (UAV)-borne laser scanning (ULS) and terrestrial laser scanning (TLS) in two German mixed forests with species typical for Central Europe. We provide the spatially overlapping, georeferenced point clouds for 12 forest plots.*

- Line 5-7 (p1) You mention the tree metrics are derived from the point clouds and measured in the field. Yet, only for half of the plots the metrics are measured in the field. This should be mentioned here.

*We specified this in the abstract (L 8f.):*
*Tree metrics were derived from the point clouds and, for half of the plots, also measured in the field.*

Introduction:

- Line 41 (p2), this sentence suggest 1491 trees were extracted from all the scanning platforms (which is not the case). So maybe start with:
  - "249 trees were extracted from both ALS, ULS and TLS. Another 1036 trees were extracted from both ALS and ULS, and xxx only from ALS."

*We now explicitly list the number of trees available for the most important combinations of datasets, see L 80-85.*
*We present 1491 trees which were identified in point clouds acquired in 12 forest plots in southwest Germany. 249 trees were extracted from all three types of point clouds, ALS, ULS and TLS. Another 1031 trees were extracted from both ALS and ULS point clouds. 1168 trees were extracted from two different ULS datasets, acquired under leaf-on conditions and leaf-off conditions (Figure 8).*

Methods:

- It would be nice to have a table with for each plot (like Table 3) the availability of the data sources (with for example also the number of trees scanned in each plot as this is not clear from the manuscript). This will help the user to choose the plots their interested in because of the availability of certain data sources and number of trees.

| Plot | ALS | ULS (leaf-on) | ULS (leaf-off) | TLS | Field measurements |
|------|-----|---------------|----------------|-----|--------------------|
| BR01 | 300 | 250 | 250 | 10 | 10 |
| BR02 | | | | | - |
| BR03 | | | | | |

| | | | | | |
|---|---|---|---|---|---|
| BR04 | | | | | - |
| BR05 | | | | | |
| BR06 | | | | | - |
| BR07 | | | | | - |
| BR08 | | | | | - |
| SP02 | | | | | - |
| KA09 | | | | | |
| KA10 | | | | | |
| KA11 | | | | | |

*We added this useful table to the manuscript in the Results section (Section 3.3, Table 6) and reference it in the Methods section as well (Section 2.3, L 197f.).*

Data:

- It would be really nice for the users if there were a file available with a table with for each tree information on the data which is collected for it. In this way the users could write a quick script to determine which trees they wants to use (e.g. only trees where of a certain species, of which there is quality 2 ALS, ULS, TLS and field data).

| tree | species | Plot | ALS q1 | … | ALS q6 | ULS (leaf-on) q1 | ULS (leaf-off) q1 | TLS Q1 | Field measurements |
|---|---|---|---|---|---|---|---|---|---|
| P17T34 | PicAbi | BR01 | FALSE | … | TRUE | FALSE | FALSE | FALSE | FALSE |

*We have created a Python software package (pytreedb) which provides a simple database interface and REST API to perform precisely this task. The object-based library is available at https://github.com/3dgeo-heidelberg/pytreedb. A web frontend allows users to query tree and tree point clouds of interest with different filters, to export data and to view maps of the query results. In the GitHub repository, we also include the dataset presented here.*
*We added the link and a small explanation to the manuscript (L 485-488).*
*We created the Python software package pytreedb (https://github.com/3dgeo-heidelberg/pytreedb) which provides a simple database interface and REST API to perform queries on a database of tree objects. pytreedb allows selecting trees using different filter, viewing maps of the selection results and exporting data in different formats (LAZ point clouds, GeoJSON files, CSV files).*

Technical corrections:

Line 7 (p1) "Our dataset may be used for the creation of 3D tree models for radiative transfer modelling or LiDAR simulation studies or to fit allometric equations between point cloud metrics and forest inventory variables" → "… creation of 3D tree models for radiative transfer modelling, LiDAR simulation studies or to fit allometric equations between point cloud metrics and forest inventory variables."

Line 301 (p19) "A visual inspection of the the wooden target …". Remove one the.

Line 306 (p19) "… and 0.3 m(Karlsruhe)." Add a comma between 0.3 m and (Karlsruhe).

Line 347 (p21) "… point clouds as fix point clouds." Shouldn't fix be fixed?

Line 431 (p29) " … and AdTree(Du et al., 2019…)." Add a comma between AdTree and (Du.

*Thank you, we made the corrections to the manuscript.*

**Referee comment 2**

**General comments**

The paper outlines a large, interesting and well-structured data set derived from ALS, ULS and TLS point clouds acquired from twelve forest plots in southwestern Germany. These LiDAR derived data are supported by traditional forest measurements. The whole dataset is available for download representing a seldomly seem large scale open access dataset encompassing LiDAR data in central Europe.

The text is well written using a high standard of (American) English in the conventional scientific style, citations are both relevant and correctly presented.

**In my opinion this manuscript can be published with minor** revisions specific comments and suggestions of technical corrections are given below.

**Specific comments**

There is a lack of information about the specific forest stands in regards to their management history, composition and structure. This is important for the end user to know in order to make comparisons within and between datasets for comparability. Within this idea, some basic stand-based metrics for each of the sample plots might be of interest for the reader (species mixture, basel area etc.) such information can be included in a supplementary data table and will aid the end user in determining which data sets (study plots) to use, maybe in conjunction with their own data.

*This is a very good idea. We added a table to the Appendix (table A1), which for each plot (and species) displays the number of trees, the basal area [$m^2 ha^{-1}$], the percentage of dead trees and the median height [m]. We reference this table in Section 2.1 Study site.*

Figure 4 nicely illustrates the effect of leaf weight on branch position in the ULS data (blue vs. pink points) – this could be highlighted in a comment for added value.

*We added this observation to the manuscript (L 328-330):*

*When comparing the TLS leaf-on point clouds with the ULS leaf-off point clouds, it is clearly visible that the branches hang down more in the summer when they are full of leaves than in the fall or winter when they are free of foliage.*

Line 226/227 "For all trees, the quality of the fit was controlled visually." Please briefly expand on this, with what threshold, what was done if the ellipse was not fitting? Visual examination is very subjective, can consistency be assured? The users of the data must understand how this was carried out.

*For transparency, we include the figures of the circle/ellipse fitting as a supplementary file. We furthermore document some details, specifically*

- a) *For which trees we had to fit a circle/ellipse to points of a slice different than 1.28 to 1.32 m above ground (different vertical position and/or thicker slice)*
- b) *For which trees the estimated DBH might be less accurate due to point cloud quality (e.g., few points on the stem) based on visual inspection*
- c) *For which trees there was a high deviation between DBH derived from the point clouds and DBH measured in the field*
- d) *For which trees ellipse fitting with least squares was carried out instead of circle fitting with RANSAC*

*We reference the supplementary file in the manuscript in L 271-273:*

*All figures of the circle/ellipse fitting are included in Supplement 1 together with a list of trees for which a) a different slice height was used, b) the visual assessment suggests low accuracy and c) TLS-derived DBH values deviate more than 10 % from the values measured in the field.*

Line 280: was there a threshold used to discard trees? More detail needed.

*We did not use a quantitative threshold, but discarded the ALS tree point clouds when they were not recognizable as a tree. This concerns only five trees, which had between 11 and 247 ALS points. We added this information to the manuscript (L 335f.).*

*This concerned only silver firs in plot SP02 (AbiAlb_SP02_01 to AbiAlb_SP02_05) which had between 11 and 247 points.*

**Technical corrections**

Splitting of words between lines – this might be a journal/template issue. While acceptable in German text But the splitting of words in English can be considered bad form. This is particularly apparent in line 152/153 across pages 9 and 10 overlap-ping.

Line 3/Line 34: Please also define UAV as "unmanned aerial vehicle" in the first instance, later defined in Line 456 as "uncrewed" this is better in a gender neutral usage.

Line 24: consider changing "The tree crowns' geometries…." to "The geometry of tree crowns can then…." Review: Weiser et al. ESSD

Line 42: Southwest → southwest, not a name of a place or region (please check the remainder of the document.

Line 54: consider changing to "the creation of 3D…."

Line 56: consider changing to "the development of…"

Line 222: slanted → leaning

Line 225: we fit an ellipse → an ellipse was fitted

Line 238: Concave → Convex

*We intentionally use concave here, so we will not change it to convex. We used two different algorithms for estimating the crown projection area, where one is based on the convex hull and one on the concave hull. The figure below illustrates the difference between these algorithms (Tree ID: FagSyl_BR02_03, ULS leaf-on point cloud acquired on 2019-08-24).*

[Figure]

[Figure]

Line 320: change to read "the good alignment *of the data sets.*"

Line 324: 1 -ha → 1 ha

Line 342: 5 iterations → five iterations (please check the rest of the document and change accordingly e.g. line 353).

Figure 10: attention to capitalisation of axis labels

*We made sure that only variable names are written in all caps and the rest is written in lower case (e.g., field inventory)*

Caption fig 10: overlap with page number

*We modified the margins so that the fitter fits well on one page.*

Line 466: like → Such as

*We appreciate your technical corrections and applied them to the manuscript.*

---

## Referee Report (RR1)

**Individual tree point clouds and tree measurements from multi-platform laser scanning in German forests**. Weiser et al.

**2nd round**

Thank you for the revised version of the paper alongside your comments on the 1st review. The manuscript has improved greatly as a result of this process. I am satisfied with the quality of the data, its description and its presentation for use by third-parties.

Re-reading the manuscript as part of this second review I have some further minor suggestions that the authors may wish to consider, please see these below. Nevertheless, I can recommend the publication of this manuscript in a close form to the revised submission.

**Technical corrections/ suggestions**

- Line 12: available for download, hosted by the PANGAEA ….
- You might consider combining tables 1 and 2 and including the information for the TLS scanner also (currently held in lines 164 onwards), this would allow direct comparison and be a bit more compact.
- Centre dashes in Table 4
- Thanks for including Table 6, this is an improvement. **Consider replacing 0 values with dashes** to aid readability, also ensure the "total" row is all on one line
- Line 374: replace "good" with "high" or similar, good is to subjective
- Line 496: Weiser et al 2021 should maybe be Weiser et al 2021c (also line 505 and line 720)
- Table A1, please add a header for relative tree count values, please also tidy up column widths and header justification. Otherwise a great addition.

---

## Author Response (AR2)

Dear Dr. Sybille Hassler, dear colleagues and reviewers,

thank you for reviewing our manuscript "Individual tree point clouds and tree measurements from multi-platform laser scanning in German forests". We incorporated the last technical corrections and suggestions. Below, we provide our responses to the comments in *italics and blue* and excerpts from the updated manuscript in *italics and orange*.

In addition to the revised manuscript, we provide a version with highlighted changes as PDF file.

Besides the requested technical corrections, we corrected the specification of the vertical field of view of the TLS acquisitions (L 142 in the version with tracked changes):

*Each scan had a vertical field of view of 100° (+60°/-40° from the horizontal plane) and …*

Furthermore, we added an additional affiliation for one of the authors, Nina Krašovec.

Yours sincerely,

Hannah Weiser

On behalf of all co-authors

**Referee comment 2**

Thank you for the revised version of the paper alongside your comments on the 1st review. The manuscript has improved greatly as a result of this process. I am satisfied with the quality of the data, its description and its presentation for use by third-parties.

Re-reading the manuscript as part of this second review I have some further minor suggestions that the authors may wish to consider, please see these below. Nevertheless, I can recommend the publication of this manuscript in a close form to the revised submission.

**Technical corrections/ suggestions**
- Line 12: available for download, hosted by the PANGAEA ….
- You might consider combining tables 1 and 2 and including the information for the TLS scanner also (currently held in lines 164 onwards), this would allow direct comparison and be a bit more compact.

  *We combined the tables according to your suggestion. Table 1 now shows scanner specifications and acquisition settings for TLS, ULS and ALS.*

- Centre dashes in Table 4
- Thanks for including Table 6, this is an improvement. **Consider replacing 0 values with dashes** to aid readability, also ensure the "total" row is all on one line

  *We kept the two rows for the number of trees per data source for all plots ( "All"; last two rows) in Table 6 (now Table 5). This is because we want to display both the sum of trees for which ULS leaf-off data is available as well as the sum for the separate times of leaf-off data acquisition (Autumn 2019, Spring 2020). Because the 503 trees in BR01 were acquired in both autumn and spring, providing just the total leaf-off number (1173) might confuse the reader, because it is not clear how this sum is formed from the numbers in the columns above. We added an explanation to the caption:*

*Note that most trees were measured from different platforms and at different times. ULS leaf-off data is available for 1173 trees. Of these, 133 trees were measured in autumn 2019, 537 trees were measured in spring 2020, and 503 trees were measured both in autumn 2019 and spring 2020.*

- Line 374: replace "good" with "high" or similar, good is to subjective
- Line 496: Weiser et al 2021 should maybe be Weiser et al 2021c (also line 505 and line 720)

*As we had to correct the year in two of the references, it is now Weiser et al. 2022b (for the PANGAEA dataset) and Weiser et al. 2022a (for the PANGAEA metadata document).*
*We fixed some other references which had a "wrong doi" (i.e. https://doi.org/https://doi.org/...) or missing author list and marked them in blue. Lastly, we updated the URLs in the references for the sensor datasheets to the documents we uploaded as supplement to our data repository.*

- Table A1, please add a header for relative tree count values, please also tidy up column widths and header justification. Otherwise a great addition.

---

## Author Response (AR3)

Dear Dr. Hassler, dear ESSD editorial and production team,

since the acceptance, I have made a few minor technical corrections listed below and uploaded a new manuscript file (and corresponding .tex and .bib files).

I hope it was okay to make these changes at this stage of the revision process.

Best regards,

Hannah Weiser

**Technical corrections:**

**Manuscript:**

- L. 15: Wang et al. 2020 → Wang et al. 2019 (now appears in the references in L 663)
- L. 241: Supplement 1 → Supplement
- L. 645: last access: 2019-05-21 → last access: 2022-04-25

**Supplement:**

- Title: beast → breast
- Added page numbers
- Added explanations for the figures on page 4